# CAUSAL PROXY MODELS FOR CONCEPT-BASED MODEL EXPLANATIONS

## ABSTRACT

Explainability methods for NLP systems encounter a version of the fundamental problem of causal inference: for a given ground-truth input text, we never truly observe the counterfactual texts necessary for isolating the causal effects of model representations on outputs. In response, many explainability methods make no use of counterfactual texts, assuming they will be unavailable. In this paper, we show that robust causal explainability methods can be created using approximate counterfactuals, which can be written by humans to approximate a specific counterfactual or simply sampled using metadata-guided heuristics. The core of our proposal is the Causal Proxy Model (CPM). A CPM explains a black-box model $\mathcal{N}$ because it is trained to have the same *actual* input/output behavior as $\mathcal{N}$ while creating neural representations that can be intervened upon to simulate the *counterfactual* input/output behavior of $\mathcal{N}$. Furthermore, we show that the best CPM for $\mathcal{N}$ performs comparably to $\mathcal{N}$ in making factual predictions, which means that the CPM can simply replace $\mathcal{N}$, leading to more explainable deployed models.

## 1 INTRODUCTION

The gold standard for explanation methods in AI should be to elucidate the *causal role* that a model's representations play in its overall behavior – to truly explain *why* the model makes the predictions it does. Causal explanation methods seek to do this by resolving the counterfactual question of what the model would do if input $X$ were changed to a relevant counterfactual version $X'$. Unfortunately, even though neural networks are fully observed, deterministic systems, we still encounter the fundamental problem of causal inference (Holland, 1986): for a given ground-truth input $X$, we never observe the counterfactual inputs $X'$ necessary for isolating the causal effects of model representations on outputs. The issue is especially pressing in domains where it is hard to synthesize approximate counterfactuals. In response to this, explanation methods typically do not explicitly train on counterfactuals at all.

In this paper, we show that robust explanation methods for NLP models can be obtained using texts approximating true counterfactuals. The heart of our proposal is the Causal Proxy Model (CPM). CPMs are trained to mimic both the factual and counterfactual behavior of a black-box model $\mathcal{N}$. We explore two different methods for training such explainers. These methods share a distillation-style objective that pushes them to mimic the factual behavior of $\mathcal{N}$, but they differ in their counterfactual objectives. The input-based method $CPM_{IN}$ appends to the factual input a new token associated with the counterfactual concept value. The hidden-state method $CPM_{HI}$ employs the Interchange Intervention Training (IIT) method of Geiger et al. (2022) to localize information about the target concept in specific hidden states. Figure 1 provides a high-level overview.

We evaluate these methods on the CEBaB benchmark for causal explanation methods (Abraham et al., 2022), which provides large numbers of original examples (restaurant reviews) with human-created counterfactuals for specific concepts (e.g., service quality), with all the texts labeled for their concept-level and text-level sentiment. We consider two types of approximate counterfactuals derived from CEBaB: texts written by humans to approximate a specific counterfactual, and texts sampled using metadata-guided heuristics. Both approximate counterfactual strategies lead to state-of-the-art performance on CEBaB for both $CPM_{IN}$ and $CPM_{HI}$.

We additionally identify two other benefits of using CPMs to explain models. First, both $CPM_{IN}$ and $CPM_{HI}$ have factual performance comparable to that of the original black-box model $\mathcal{N}$ and can explain their own behavior extremely well. Thus, the CPM for $\mathcal{N}$ can actually replace $\mathcal{N}$, leading to more explainable deployed models. Second, $CPM_{HI}$ models localize concept-level information in

their hidden representations, which makes their behavior on specific inputs very easy to explain. We illustrate this using Path Integrated Gradients (Sundararajan et al., 2017), which we adapt to allow input-level attributions to be mediated by the intermediate states that were targeted for localization. Thus, while both $CPM_{IN}$ and $CPM_{HI}$ are comparable as explanation methods according to CEBaB, the qualitative insights afforded by $CPM_{HI}$ models may given them the edge when it comes to explanations.

## 2 RELATED WORK

Understanding model behavior serves many goals for large-scale AI systems, including transparency (Kim, 2015; Lipton, 2018; Pearl, 2019; Ehsan et al., 2021), trustworthiness (Ribeiro et al., 2016; Guidotti et al., 2018; Jacovi & Goldberg, 2020; Jakesch et al., 2019), safety (Amodei et al., 2016; Otte, 2013), and fairness (Hardt et al., 2016; Kleinberg et al., 2017; Goodman & Flaxman, 2017; Mehrabi et al., 2021). With CPMs, our goal is to achieve explanations that are causally motivated and concept-based, and so we concentrate here on relating existing methods to these two goals.

Feature attribution methods estimate the importance of features, generally by inspecting learned weights directly or by perturbing features and studying the effects this has on model behavior (Molnar, 2020; Ribeiro et al., 2016). Gradient-based feature attribution methods extend this general mode of explanation to the hidden representations in deep networks (Zeiler & Fergus, 2014; Springenberg et al., 2014; Binder et al., 2016; Shrikumar et al., 2017; Sundararajan et al., 2017). Concept Activation Vectors (CAVs; Kim et al. 2018; Yeh et al. 2020) can also be considered feature attribution methods, as they probe for semantically meaningful directions in the model's internal representations and use these to estimate the importance of concepts on the model predictions. While some methods in this space do have causal interpretations (e.g., Sundararajan et al. 2017; Yeh et al. 2020), most do not. In addition, most of these methods offer explanations in terms of specific (sets of) features/neurons. (Methods based on CAVs operate directly in terms of more abstract concepts.)

Intervention-based methods study model representations by modifying them in systematic ways and observing the resulting model behavior. These methods are generally causally motivated and allow for concept-based explanations. Examples of methods in this space include causal mediation analysis (Vig et al., 2020; De Cao et al., 2021; Ban et al., 2022), causal effect estimation (Feder et al., 2020; Elazar et al., 2021; Abraham et al., 2022; Lovering & Pavlick, 2022), tensor product decomposition (Soulos et al., 2020), and causal abstraction analysis (Geiger et al., 2020; 2021). CPMs are most closely related to the method of IIT (Geiger et al., 2021), which extends causal abstraction analysis to optimization.

Probing is another important class of explanation method. Traditional probes do not intervene on the target model, but rather only seek to find information in it via supervised models (Conneau et al., 2018; Tenney et al., 2019) or unsupervised models (Clark et al., 2019; Manning et al., 2020; Saphra & Lopez, 2019). Probes can identify concept-based information, but they cannot offer guarantees that probed information is relevant for model behavior (Geiger et al., 2021). For causal guarantees, it is likely that some kind of intervention is required. For example, Elazar et al. (2021) and Feder et al. (2020) remove information from model representations to estimate the causal role of that information. Our CPMs employ a similar set of guiding ideas but are not limited to removing information.

Counterfactual explanation methods aim to explain model behavior by providing a counterfactual example that changes the model behavior (Goyal et al., 2019; Verma et al., 2020; Wu et al., 2021). Counterfactual explanation methods are inherently causal. If they can provide counterfactual examples with regard to specific concepts, they are also concept-based.

Some explanation methods train a model making explicit use of intermediate variables representing concepts. Manipulating these intermediate variables at inference time yields causal concept-based model explanations (Koh et al., 2020; Künzel et al., 2019).

Evaluating methods in this space has been a persistent challenge. In prior literature, explanation methods have often been evaluated against synthetic datasets (Feder et al., 2020; Yeh et al., 2020). In response, Abraham et al. (2022) introduced the CEBaB dataset, which provides a human-validated concept-based dataset to truthfully evaluate different causal concept-based model explanation methods. Our primary evaluations are conducted on CEBaB.

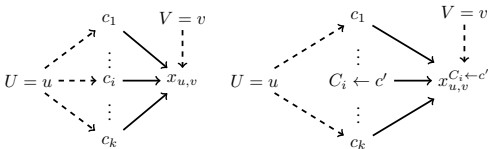

(a) A structural causal model leading to an actual text $x_{u,v}$ and its counterfactual text $x_{u,v}^{C_i \leftarrow c'}$. $U$ is an exogenous variable over experiences, $c_1, \ldots, c_k$ are mediating concepts, and $V$ is an exogenous variable capturing the writing (and star-rating) experience. At right, we create a counterfactual in which concept $C_i$ takes on a different value. Unfortunately, we cannot truly create such counterfactual situations and so we never observe pairs of texts like these. Thus, we must rely on approximate counterfactuals.

Let $x_{u,v}$ be a text written in situation $(u, v)$:

**Human-created** $\tilde{x}_{u,v}^{C_i \leftarrow c'}$

Crowdworker edit of $x_{u,v}$ to express that $C_i$ had value $c'$, seeking to keep all else constant.

**Metadata-sampled** $\tilde{x}_{u,v}^{C_i \leftarrow c'}$

Sampled text expressing that $C_i$ has value $c'$ but agreeing with $x_{u,v}$ on all other concepts.

(b) Approximate counterfactuals. In the human-created strategy, humans revise an attested text to try to express a particular counterfactual, seeking to simulate a causal intervention. In the metadata-sampled strategy, we find a separate text that aligns with the original for the value $u$ insofar as it expresses all the same concepts except for the target concept $C_i$.

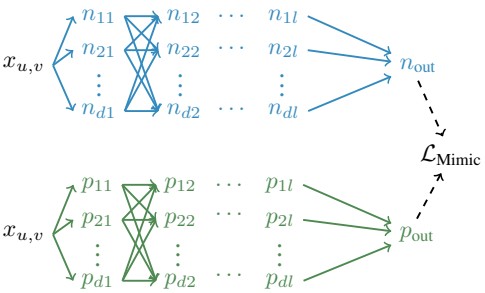

(c) $\mathcal{L}_{\text{Mimic}}$: All CPMs (bottom) are trained to mimic the behavior of the neural model $\mathcal{N}$ to be explained (top) for all factual inputs $x_{u,v}$.

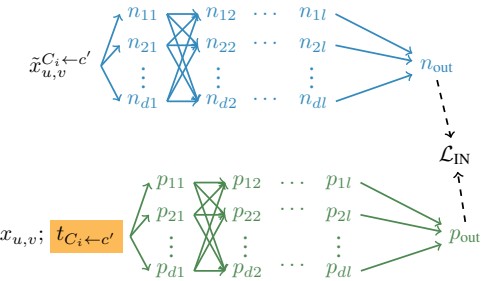

(d) $\mathcal{L}_{\text{IN}}$: Examples $x_{u,v}$ and $\tilde{x}_{u,v}^{C_i \leftarrow c'}$ are an approximate counterfactual pair. The CPM is given $x_{u,v}$ augmented with a special token $t_{C_i \leftarrow c'}$ and trained to mimic the target model $\mathcal{N}$ when its input is $\tilde{x}_{u,v}^{C_i \leftarrow c'}$.

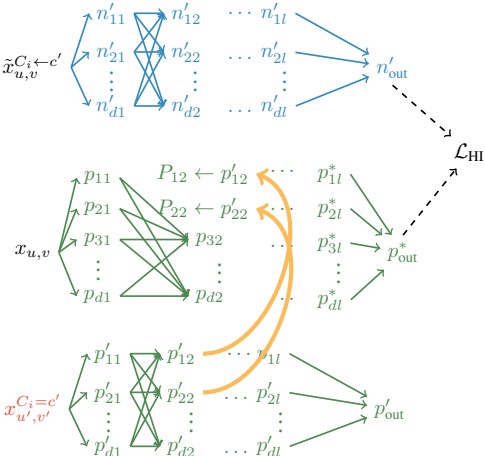

(e) $\mathcal{L}_{\text{HI}}$: Examples $x_{u,v}$ and $\tilde{x}_{u,v}^{C_i \leftarrow c'}$ are an approximate counterfactual pair. The CPM (middle) is given input $x_{u,v}$. The objective is for it to mimic $\mathcal{N}$ (top) given $\tilde{x}_{u,v}^{C_i \leftarrow c'}$, but under the intervention in which specific internal states are changed to those that the CPM computes for input $x_{u',v'}^{C_i = c'}$ (bottom), which is a distinct example that is sampled with the only criteria being that it express $C_i = c'$. The effect of this intervention is to localize information about concept $C_i$ at the intervention site, since the only indication the CPM gets about $C_i \leftarrow c'$ is via the intervention.

Figure 1: Causal Proxy Model (CPM) summary. Every CPM for model $\mathcal{N}$ is trained to mimic the factual behavior of $\mathcal{N}$ ($\mathcal{L}_{\text{Mimic}}$). For CPM$_{\text{IN}}$, the counterfactual objective is $\mathcal{L}_{\text{IN}}$. For CPM$_{\text{HI}}$, the counterfactual objective is $\mathcal{L}_{\text{HI}}$.

# 3 CAUSAL PROXY MODEL (CPM)

Causal Proxy Models (CPMs) are causal concept-based explanation methods. Given a factual input $x_{u,v}$ and a description of a concept intervention $C_i \leftarrow c'$, they estimate the effect of the intervention on model output. The present section introduces our two core CPM variants in detail. We concentrate here on introducing the structure of these models and their objectives, and we save discussion of associated metrics for explanation methods for Section 4.

**A Structural Causal Model** Our discussion is grounded in the causal model depicted in Figure 1a, which aligns well with the CEBaB benchmark. Two exogenous variables $U$ and $V$ together represent the complete state of the world and generate some textual data $X$. The effect of exogenous variable $U$ on the data $X$ is completely mediated by a set of intermediate variables $C_1, C_2 \ldots, C_k$, which we refer to as *concepts*. Therefore, we can think of $U$ as the part of the world that gives rise to these concepts $\{C\}_1^k$.

Using this causal model, we can describe counterfactual data – data that arose under a counterfactual state of the world (right diagram in Figure 1a). Our factual text is $x_{u,v}$, and we use $x_{u,v}^{C_i \leftarrow c'}$ for the counterfactual text obtained by intervening on concept $C_i$ to set its value to $c'$. The counterfactual $x_{u,v}^{C_i \leftarrow c'}$ describes the output when the value of $C_i$ is set to $c'$, all else being held equal.

**Approximate Counterfactuals** Unfortunately, pairs like $(x_{u,v}, x_{u,v}^{C_i \leftarrow c'})$ are never observed, and thus we need strategies for creating approximate counterfactuals $\tilde{x}_{u,v}^{C_i \leftarrow c'}$. Figure 1b describes the two strategies we use in this paper. In the human-created strategy, we rely on a crowdworker to edit $x_{u,v}$ to achieve a particular counterfactual goal – say, making the evaluation of the restaurant's food negative. CEBaB contains an abundance of such pairs $(x_{u,v}, \tilde{x}_{u,v}^{C_i \leftarrow c'})$. However, CEBaB is unusual in having so many human-created approximate counterfactuals, so we also explore a simpler strategy in which $\tilde{x}_{u,v}^{C_i \leftarrow c'}$ is sampled with the requirement that it match $x_{u,v}$ on all concepts but sets $C_i$ to $c'$. This strategy is supported in many real-world datasets – for example, the OpenTable reviews underlying CEBaB all have the needed metadata (Abraham et al., 2022).

**CPM$_{\text{IN}}$: Input-based CPM** Given a dataset of approximate counterfactual pairs $(x_{u,v}, \tilde{x}_{u,v}^{C_i \leftarrow c'})$ and a black-box model $\mathcal{N}$, we train a new CPM$_{\text{IN}}$ model $\mathcal{P}$ with a counterfactual objective as:

$$\mathcal{L}_{\text{IN}} = \text{CE}_{\text{S}}\big(\mathcal{N}(\tilde{x}_{u,v}^{C_i \leftarrow c'}), \mathcal{P}(x_{u,v}; t_{C_i \leftarrow c'})\big) \tag{1}$$

where $x_{u,v}; t_{C_i \leftarrow c'}$ in Eqn. 1 denotes the concatenation of the factual input and a randomly initialized learnable token embedding $t_{C_i \leftarrow c'}$ describing the intervention $C_i \leftarrow c'$. $\text{CE}_{\text{S}}$ represents the smoothed cross-entropy loss (Hinton et al., 2015), measuring the divergence between the output logits of both models. The objective in Eqn. 1 pushes $\mathcal{P}$ to predict the counterfactual behavior of $\mathcal{N}$ when a descriptor of the intervention is given (Figure 1d).[1]

At inference time, approximate counterfactuals are inaccessible. To explain model $\mathcal{N}$, we append the trained token embedding $t_{C_i \leftarrow c'}$ to a factual input, upon which $\mathcal{P}$ predicts a counterfactual output for this input, used to estimate the counterfactual behavior of $\mathcal{N}$ under this intervention.

**CPM$_{\text{HI}}$: Hidden-state CPM** Our CPM$_{\text{HI}}$ models are trained on the same data and with the same set of goals as CPM$_{\text{IN}}$, to mimic both the factual and counterfactual behavior of $\mathcal{N}$. The key difference is how the information about the intervention $C_i \leftarrow c'$ is exposed to the model. Specifically, we adapt Interchange Intervention Training (Geiger et al., 2022) to train our CPM$_{\text{HI}}$ models for concept-based model explanation.

A conventional intervention on a hidden representation $H$ of a neural network $\mathcal{N}$ fixes the value of the representation $H$ to a constant. In an interchange intervention, we instead fix $H$ to the value it would have been when processing a separate source input $s$. The result of the interchange intervention is a new model. Formally, we describe this new model as $\mathcal{N}_{H \leftarrow H_s}$, where $\leftarrow$ is the conventional intervention operator and $H_s$ is the value of hidden representation $H$ when processing input $s$.

Given a dataset of approximate counterfactual input pairs $(x_{u,v}, \tilde{x}_{u,v}^{C_i \leftarrow c'})$ and a black-box model $\mathcal{N}$, we train a new CPM$_{\text{HI}}$ model $\mathcal{P}$ with the following counterfactual objective:

$$\mathcal{L}_{\text{HI}} = \text{CE}_{\text{S}}\big(\mathcal{N}(\tilde{x}_{u,v}^{C_i \leftarrow c'}), \mathcal{P}_{H^{C_i} \leftarrow H_s^{C_i}}(x_{u,v})\big) \tag{2}$$

Here $H^{C_i}$ are hidden states designated for concept $C_i$. In essence, we train $\mathcal{P}$ to fully mediate the effect of intervening on $C_i$ in the hidden representation $H^{C_i}$. The source input $s$ is any input $x_{u',v'}^{C_i = c'}$ that has $C_i = c'$. As $\mathcal{P}$ only receives information about the concept-level intervention $C_i \leftarrow c'$ via the

---

[1]Our objective is regard to a single approximate counterfactual pair for the sake of clarity. At train-time, we aggregate the objective over all considered training pairs. We take $C_i$ to always represent the intervened-upon concept. The weights of $\mathcal{N}$ are frozen.

interchange intervention $H^{C_i} \leftarrow H^{C_i}_s$, the model is forced to store all causally relevant information with regard to $C_i$ in the corresponding hidden representation. This process is described in Figure 1e.

In the ideal situation, the source input $x^{C_i=c'}_{u',v'}$ and $x_{u,v}$ share the same value only for $C_i$ and differ on all others, so that the counterfactual signal needed for localization is pure. However, we do not insist on this when we sample. In addition, we allow *null effect* pairs in which $x_{u,v}$ and $\tilde{x}^{C_i \leftarrow c'}_{u,v}$ are identical. For additional details on this sampling procedure, see Appendix A.2.

At inference time, approximate counterfactuals are inaccessible, as before. To explain model $\mathcal{N}$ with regard to intervention $C_i \leftarrow c'$, we manipulate the internal states of model $\mathcal{P}$ by intervening on the localized representation $H^{C_i}$ for concept $C_i$. To achieve this, we sample a source input $x^{C_i=c'}_{u',v'}$ from the train set as any input $x$ that has $C_i = c'$ to derive $H^{C_i}_s$.

**Training Objectives** We include another distillation objective to predict the same output as $\mathcal{N}$ under conventional circumstances as $\mathcal{L}_{\text{Mimic}} = \text{CE}_\text{S}\big(\mathcal{N}(x_{u,v}), \mathcal{P}(x_{u,v})\big)$. The overall training objective for our models can be written as $\mathcal{L} = \lambda_1 \mathcal{L}_{\text{Mimic}} + \lambda_2 \mathcal{L}_{\text{Counterfactual}}$ where $\mathcal{L}_{\text{Counterfactual}}$ can be either $\mathcal{L}_{\text{IN}}$ or $\mathcal{L}_{\text{HI}}$, and we set $\lambda_1, \lambda_2$ as 1.0 and 3.0 for simplicity.

## 4 EXPERIMENT SETUP

### 4.1 CAUSAL ESTIMATION-BASED BENCHMARK (CEBaB)

CEBaB (Abraham et al., 2022) is a large benchmark of high-quality, labeled approximate counterfactuals for the task of sentiment analysis on restaurant reviews. The benchmark was created starting from a set of 2,299 original restaurant reviews from OpenTable. For each of these original reviews, approximate counterfactual examples were written by human annotators; the annotators were tasked to edit the original text to reflect a specific intervention, like 'change the food evaluation from negative to positive' or 'change the service evaluation from positive to unknown'. In this way, the original reviews were expanded with approximate counterfactuals to a total of 15,089 texts. The groups of originals and corresponding approximate counterfactuals are partitioned over train, dev, and test sets. The pairs in the development and test set are used to benchmark explanation methods.

Each text in CEBaB was labeled by five crowdworkers with a 5-star sentiment score. In addition, each text was annotated at the concept level for four mediating concepts $\{C_{\text{ambiance}}, C_{\text{food}}, C_{\text{noise}},$ and $C_{\text{service}}\}$, using the labels $\{$negative, unknown, positive$\}$, again with five crowdworkers annotating each concept-level label. We refer to Appendix A.1 and Abraham et al. 2022 for additional details.

As discussed above (Section 3 and Figure 1b), we consider two sources of approximate counterfactuals using CEBaB. For human-created counterfactuals, we use the edited restaurant reviews of the train set. For metadata-sampled counterfactuals, we sample factual inputs from the train set that have the desired combination of mediating concepts. Using all the human-created edits leads to 19,684 training pairs of factuals and corresponding approximate counterfactuals. Sampling counterfactuals leads to 74,574 pairs. We use these approximate counterfactuals to train explanation methods. Appendix A.2 provides more information about our pairing process.

### 4.2 EVALUATION METRICS

Much of the value of a benchmark like CEBaB derives from its support for directly calculating the Estimated Individual Causal Concept Effect ($\widehat{\text{ICaCE}}_\mathcal{N}$) for a model $\mathcal{N}$ given a human-generated approximate counterfactual pair $(x_{u,v}, \tilde{x}^{C_i \leftarrow c'}_{u,v})$:

$$\widehat{\text{ICaCE}}_\mathcal{N}(x_{u,v}, \tilde{x}^{C_i \leftarrow c'}_{u,v}) = \mathcal{N}(\tilde{x}^{C_i \leftarrow c'}_{u,v}) - \mathcal{N}(x_{u,v}) \tag{3}$$

This is simply the difference between the vectors of output scores for the two examples.

We do not expect to have pairs $(x_{u,v}, \tilde{x}^{C_i \leftarrow c'}_{u,v})$ at inference time, and this is what drives the development of explanation methods $\mathcal{E}_\mathcal{N}$ that *estimate* this quantity using only a factual input $x_{u,v}$ and a description of the intervention $C_i \leftarrow c'$. To benchmark such methods, we follow Abraham et al.

(2022) in using the ICaCE-Error:

$$\text{ICaCE-Error}_{\mathcal{N}}^{\mathcal{D}}(\mathcal{E}) = \frac{1}{|\mathcal{D}|} \sum_{(x_{u,v}, \tilde{x}_{u,v}^{C_i \leftarrow c'}) \in \mathcal{D}} \text{Dist}\big(\widehat{\text{ICaCE}}_{\mathcal{N}}((x_{u,v}, \tilde{x}_{u,v}^{C_i \leftarrow c'})), \mathcal{E}_{\mathcal{N}}(x_{u,v}; C_i \leftarrow c')\big) \quad (4)$$

Here, we assume that $\mathcal{D}$ is a dataset consisting entirely of approximate counterfactual pairs $(x_{u,v}, \tilde{x}_{u,v}^{C_i \leftarrow c'})$. Dist measures the distance between the $\widehat{\text{ICaCE}}_{\mathcal{N}}$ for the model $\mathcal{N}$ and the effect predicted by the explanation method. Abraham et al. (2022) consider three values for Dist: L2, which captures both direction and magnitude; Cosine distance, which captures the direction of effects but not their magnitude; and NormDiff (absolute difference of L2 norms), which captures magnitude but not direction. We report all three metrics.

### 4.3 BASELINE METHODS

**BEST$_{\text{CEBaB}}$** We compare our results with the best results obtained on the CEBaB benchmark. Crucially, BEST$_{\text{CEBaB}}$ consists aggregated best results from a set of methods including CONEXP (Goyal et al., 2020), TCAV (Kim et al., 2018), ConceptSHAP (Yeh et al., 2020), INLP (Ravfogel et al., 2020), CausaLM (Feder et al., 2020), and S-Learner (Künzel et al., 2019).

**S-Learner** Our version of S-Learner (Künzel et al., 2019) learns to mimic the factual behavior of black-box model $\mathcal{N}$ while making the intermediate concepts explicit. Given a factual input, a finetuned model $\mathcal{B}$ is trained to predict concept label for each concept as an aspect-based sentiment classification task. Then, a logistic regression model $\text{LR}_{\mathcal{N}}$ is trained to map these intermediate concept values to the factual output of black-box model $\mathcal{N}$, under the following objective.

$$\mathcal{L}_{\text{Mimic}}^{\text{S},\mathcal{B}} = \text{CE}_{\text{S}}\big(\mathcal{N}(x_{u,v}), \text{LR}_{\mathcal{N}}(\mathcal{B}(x_{u,v}))\big) \quad (5)$$

By intervening on the intermediate predicted concept values at inference-time, we can hope to simulate the counterfactual behavior of $\mathcal{N}$:

$$\mathcal{E}_{\mathcal{N}}^{\text{S},\mathcal{B}}(x_{u,v}; C_i \leftarrow c') = \text{LR}_{\mathcal{N}}((\mathcal{B}(x_{u,v}))_{C_i \leftarrow c'}) - \text{LR}_{\mathcal{N}}(\mathcal{B}(x_{u,v})) \quad (6)$$

When using S-Learner in conjunction with approximate counterfactual inputs at train-time, we simply add this counterfactual data on top of the observational data that is typically used to train S-Learner.

**GPT-3** Large language models such as GPT-3 (175B) have shown extraordinary power in terms of in-context learning (Brown et al., 2020).[2] We use GPT-3 to generate a new approximate counterfactual at inference time given a factual input and a descriptor of the intervention. This generated counterfactual is directly used to estimate the change in model behavior:

$$\mathcal{E}_{\mathcal{N}}^{\text{GPT-3}}(x_{u,v}; C_i \leftarrow c') = \mathcal{N}(\text{GPT-3}(x_{u,v}; C_i \leftarrow c')) - \mathcal{N}(x_{u,v}) \quad (7)$$

where $\text{GPT-3}(x_{u,v}; C_i \leftarrow c')$ represents the GPT-3 generated counterfactual edits. We prompt GPT-3 with demonstrations containing approximate counterfactual inputs. Full details on how these prompts are constructed can be found in Appendix A.7.

### 4.4 CAUSAL PROXY MODELS

We train CPMs for the publicly available models released for CEBaB, fine-tuned as five-way sentiment classifiers on the factual data. This includes four model architectures: bert-base-uncased (BERT; Devlin et al. 2019), RoBERTa-base (RoBERTa; Liu et al. 2019), GPT-2 (GPT-2; Radford et al. 2019), and LSTM+GloVe (LSTM; Hochreiter & Schmidhuber 1997; Pennington et al. 2014). All Transformer-based models (Vaswani et al., 2017) have 12 Transformer layers. Before training, each CPM model is initialized with the architecture and weights of the black-box model we aim to explain. Thus, the CPMs are rooted in the factual behavior of $\mathcal{N}$ from the start. We include details about our setup in Appendix A.3.

The inference time comparisons for these models are as follows, where $\mathcal{P}$ in Eqn. 8 and Eqn. 9 refers to the CPM model trained under CPM$_{\text{IN}}$ and CPM$_{\text{HI}}$ objectives, respectively:

$$\mathcal{E}_{\mathcal{N}}^{\text{CPM}_{\text{IN}}}(x_{u,v}; C_i \leftarrow c') = \mathcal{P}(x_{u,v}; t_{C_i \leftarrow c'}) - \mathcal{N}(x_{u,v}) \quad (8)$$

$$\mathcal{E}_{\mathcal{N}}^{\text{CPM}_{\text{HI}}}(x_{u,v}; C_i \leftarrow c') = \mathcal{P}_{H^{C_i} \leftarrow H_s^{C_i}}(x_{u,v}) - \mathcal{N}(x_{u,v}) \quad (9)$$

---

[2]We use the largest davinci model publicly available at https://beta.openai.com/playground.

| Model | Metric | no counterfactuals | | sampled counterfactuals | | | | human-created counterfactuals | | | |
|---|---|---|---|---|---|---|---|---|---|---|---|
| | | $\text{BEST}_{\text{CEBaB}}$ | S-Learner | S-Learner | GPT-3 | **(ours)** $\text{CPM}_{\text{IN}}$ | **(ours)** $\text{CPM}_{\text{HI}}$ | S-Learner | GPT-3 | **(ours)** $\text{CPM}_{\text{IN}}$ | **(ours)** $\text{CPM}_{\text{HI}}$ |
| BERT | L2 | 0.74 (.02) | 0.74 (.02) | 0.74 (.02) | 0.71 (.01) | 0.63 (.01) | **0.60** (.01) | 0.73 (.02) | **0.45** (.01) | **0.45** (.02) | **0.45** (.03) |
| | Cosine | 0.59 (.03) | 0.63 (.01) | 0.63 (.01) | 0.51 (.00) | 0.46 (.00) | **0.45** (.00) | 0.60 (.01) | 0.36 (.00) | **0.35** (.00) | 0.36 (.04) |
| | NormDiff | 0.44 (.01) | 0.54 (.02) | 0.53 (.02) | **0.35** (.01) | 0.39 (.01) | 0.38 (.00) | 0.52 (.02) | 0.25 (.00) | **0.24** (.01) | 0.27 (.01) |
| RoBERTa | L2 | 0.78 (.01) | 0.78 (.01) | 0.78 (.00) | 0.74 (.01) | **0.66** (.01) | 0.67 (.02) | 0.77 (.00) | **0.46** (.01) | **0.46** (.01) | 0.47 (.03) |
| | Cosine | 0.58 (.01) | 0.64 (.01) | 0.65 (.01) | 0.53 (.01) | **0.46** (.00) | 0.47 (.00) | 0.63 (.01) | 0.39 (.00) | **0.38** (.01) | 0.39 (.03) |
| | NormDiff | 0.45 (.00) | 0.59 (.01) | 0.58 (.00) | **0.36** (.00) | 0.42 (.01) | 0.45 (.03) | 0.56 (.00) | 0.28 (.01) | **0.26** (.01) | 0.29 (.05) |
| GPT-2 | L2 | 0.60 (.02) | 0.60 (.02) | 0.61 (.01) | 0.65 (.01) | 0.55 (.01) | **0.51** (.01) | 0.61 (.01) | 0.43 (.01) | **0.41** (.01) | **0.41** (.04) |
| | Cosine | 0.59 (.01) | 0.59 (.01) | 0.59 (.01) | 0.52 (.00) | 0.47 (.01) | **0.46** (.00) | 0.59 (.01) | 0.40 (.00) | **0.37** (.01) | 0.39 (.05) |
| | NormDiff | 0.40 (.01) | 0.40 (.01) | 0.41 (.01) | 0.34 (.00) | 0.32 (.01) | **0.30** (.00) | 0.40 (.01) | 0.24 (.01) | **0.23** (.01) | 0.27 (.05) |
| LSTM | L2 | 0.73 (.01) | 0.73 (.01) | 0.73 (.01) | 0.76 (.00) | 0.66 (.01) | **0.64** (.02) | 0.72 (.00) | **0.49** (.00) | 0.52 (.00) | 0.54 (.01) |
| | Cosine | 0.64 (.01) | 0.64 (.01) | 0.64 (.01) | 0.57 (.01) | **0.50** (.00) | **0.50** (.01) | 0.63 (.01) | **0.44** (.00) | 0.45 (.01) | 0.46 (.00) |
| | NormDiff | 0.50 (.01) | 0.53 (.01) | 0.53 (.00) | **0.41** (.00) | 0.42 (.00) | **0.41** (.01) | 0.54 (.00) | **0.30** (.00) | 0.34 (.01) | 0.36 (.00) |

Table 1: CEBaB scores measured in three different metrics on the test set for four different model architectures as a five-class sentiment classification task. **Lower is better**. Results averaged over three distinct seeds, standard deviations in parentheses. The metrics are described in Section 4. Best averaged result is bolded (including ties) per approximate counterfactual creation strategy.

Here, $s$ is a source input with $C_i = c'$, and $H^{C_i}$ is the neural representation associated with $C_i$ which takes value $H_s^{C_i}$ on the source input $s$. As $H^{C_i}$, we use the representation of the [CLS] token. Specifically, for BERT we use slices of width 192 taken from the 1st intermediate token of the 10th layer. For RoBERTa, we use the 8th layer instead. For GPT-2, we pick the final token of the 12th layer, again with slice width of 192. For LSTM, we consider slices of the attention-gated sentence embedding with width 64. Appendix A.5 studies the impact of intervention location and size.

Following the guidance on IIT given by Geiger et al. (2022), we train $\text{CPM}_{\text{HI}}$ with an additional multi-task objective as $\mathcal{L}_{\text{Multi}} = \sum_{C_i \in C} \text{CE}(\text{MLP}(H_x^{C_i}), c)$ where probe is parameterized by a multilayer perceptron MLP, and $H_x^{C_i}$ is the value of hidden representation for the concept $C_i$ when processing input $x$ with a concept label of $c$ for $C_i$.

## 5 RESULTS

We first benchmark both CPM variants and our baseline methods on CEBaB. We show that the CPMs achieve state-of-the-art performance, for both types of approximate counterfactuals used during training (Section 5.1). Given the good factual performance achieved by CPMs, we subsequently investigate whether CPMs can be deployed both as predictor and explanation method at the same time (Section 5.2) and find that they can. Finally, we show that the localized representations of $\text{CPM}_{\text{HI}}$ give rise to concept-aware feature attributions (Section 5.3). Our supplementary materials report on detailed ablation studies and explore the potential of our methods for model debiasing.

### 5.1 CEBaB PERFORMANCE

Table 1 presents our main results. The results are grouped per approximate counterfactual type used during training. Both $\text{CPM}_{\text{IN}}$ and $\text{CPM}_{\text{HI}}$ beat $\text{BEST}_{\text{CEBaB}}$ in every evaluation setting by a large margin, establishing state-of-the-art explanation performance. Interestingly, $\text{CPM}_{\text{HI}}$ seems to slightly outperform $\text{CPM}_{\text{IN}}$ using *sampled* approximate counterfactuals, while slightly underperforming $\text{CPM}_{\text{IN}}$ on *human-created* approximate counterfactuals. Appendix A.6 reports on ablation studies that indicate that, for $\text{CPM}_{\text{HI}}$, this state-of-the-art performance is primarily driven by the role of IIT in localizing concepts.

S-Learner, one of the best individual explainers from the original CEBaB paper (Abraham et al., 2022), shows only a marginal improvement when naively incorporating *sampled* and *human-created* counterfactuals during training over using *no counterfactuals*. This indicates that the large performance gains achieved by our CPMs over previous explainers are most likely due to the explicit use of a counterfactual training signal, and not primarily due to the addition of extra (counterfactual) data.

GPT-3 occasionally performs on-par with our CPMs, generally only slightly underperforming our best explainer on *human-created counterfactuals*, while being significantly worse on *sampled counter-*

| Model | Metric | sampled counterfactuals | | human-created counterfactuals | |
|---|---|---|---|---|---|
| | | $CPM_{IN}$ | $CPM_{HI}$ | $CPM_{IN}$ | $CPM_{HI}$ |
| BERT | L2 | 0.63 (.01) | 0.52 (.04) | 0.42 (.02) | 0.38 (.03) |
| | Cosine | 0.46 (.00) | 0.45 (.01) | 0.34 (.02) | 0.30 (.06) |
| | NormDiff | 0.39 (.01) | 0.33 (.02) | 0.23 (.01) | 0.22 (.05) |
| RoBERTa | L2 | 0.66 (.01) | 0.63 (.04) | 0.40 (.01) | 0.37 (.04) |
| | Cosine | 0.46 (.00) | 0.48 (.01) | 0.33 (.01) | 0.29 (.04) |
| | NormDiff | 0.42 (.01) | 0.42 (.05) | 0.21 (.01) | 0.23 (.05) |
| GPT-2 | L2 | 0.55 (.01) | 0.41 (.03) | 0.38 (.01) | 0.36 (.04) |
| | Cosine | 0.47 (.01) | 0.39 (.02) | 0.37 (.01) | 0.35 (.05) |
| | NormDiff | 0.32 (.01) | 0.25 (.02) | 0.22 (.01) | 0.24 (.05) |
| LSTM | L2 | 0.66 (.01) | 0.41 (.01) | 0.46 (.00) | 0.42 (.01) |
| | Cosine | 0.50 (.00) | 0.42 (.02) | 0.50 (.02) | 0.40 (.01) |
| | NormDiff | 0.42 (.00) | 0.25 (.00) | 0.31 (.00) | 0.28 (.02) |

Table 3: Self-explanation CEBaB scores measured in three different metrics on the test set for four different model architectures as a five-class sentiment classification task. **Lower is better**. Results averaged over three distinct seeds, standard deviations in parentheses.

| Model | Black-box | sampled counterfactuals | | human-created counterfactuals | |
|---|---|---|---|---|---|
| | | $CPM_{IN}$ | $CPM_{HI}$ | $CPM_{IN}$ | $CPM_{HI}$ |
| BERT | 0.70 (.01) | 0.70 (.00) | 0.67 (.02) | 0.70 (.01) | 0.69 (.01) |
| RoBERTa | 0.70 (.00) | 0.70 (.00) | 0.69 (.01) | 0.71 (.01) | 0.71 (.00) |
| GPT-2 | 0.65 (.00) | 0.65 (.00) | 0.67 (.01) | 0.66 (.01) | 0.68 (.00) |
| LSTM | 0.60 (.01) | 0.60 (.01) | 0.56 (.00) | 0.54 (.00) | 0.59 (.01) |

Table 2: Task performance measured as `Macro-F1` score on the test set. Results averaged over three distinct seeds; standard deviations in parentheses.

*factuals*. While the GPT-3 explainer also explicitly uses approximate counterfactual data, the results indicate that our proposed counterfactual mimic objectives give better results. The better performance of CPMs when considering *sampled counterfactuals* over GPT-3 shows that our approach is more robust to the quality of the approximate counterfactuals used. While the GPT-3 explainer is easy to set up (no training required), it might not be suitable for some explanation applications regardless of performance, due to the latency and cost involved in querying the GPT-3 API.

Across the board, explainers trained with *human-created* counterfactuals are better than those trained with *sampled* counterfactuals. This shows that the performance of explanation methods depends on the quality of the approximate counterfactual training data. While human counterfactuals give excellent performance, they may be expensive to create. Sampled counterfactuals are cheaper if the relevant metadata is available. Thus, under budgetary constraints, sampled counterfactuals may be more efficient.

Finally, $CPM_{IN}$ is conceptually the simpler of the two CPM variants. However, we discuss in Section 5.3 how the localized representations of $CPM_{HI}$ lead to additional explainability benefits.

## 5.2  SELF-EXPLANATION WITH CPM

As outlined in Section 3, CPMs learn to mimic both the factual and counterfactual behavior of the black-box models they are explaining. We show in Table 2 that our CPMs achieve a factual `Macro-F1` score comparable to the black-box finetuned models.

We investigate if we can simply replace the black-box model with our CPM and use the CPM both as factual predictor and counterfactual explainer. To answer this questions, we measure the self-explanation performance of CPMs by simply replacing the black-box model $\mathcal{N}$ in Eqn. 4 with our factual CPM predictions at inference time.

Table 3 reports these results. We find that both $CPM_{IN}$ and $CPM_{HI}$ achieve better self-explanation performance compared to providing explanations for another black-box model. Furthermore, $CPM_{HI}$ provides better self-explanation than $CPM_{IN}$, suggesting our interchange intervention procedure leads the model to localize concept-based information in hidden representations. This shows that CPMs may be viable as replacements for their black-box counterpart, since they provide similar task performance while providing faithful counterfactual explanations of both the black-box model and themselves.

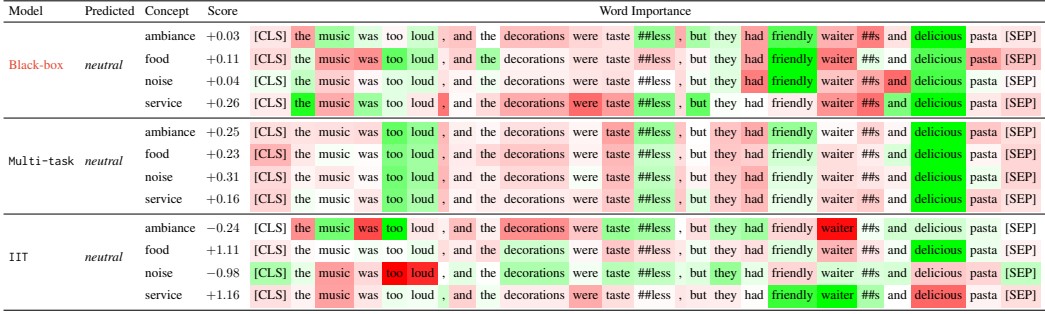

Table 4: Visualizations of word importance scores using Integrated Gradient (IG) by restricting gradient flow through the corresponding intervention site of the targeted concept. Our target class pools *positive* and *very positive*. Individual word importance is the sum of neuron-level importance scores for each input, normalized to [ $-1$ , $+1$ ]. $-1$ means the word contributes the most negatively to predicting the target class (red); $+1$ means the word contributes the most positively (green).

## 5.3   CONCEPT-AWARE FEATURE ATTRIBUTION WITH CPM$_{\text{HI}}$

We have shown that CPM$_{\text{HI}}$ provides trustworthy explanations (Section 5.1). We now investigate whether CPM$_{\text{HI}}$ learns representations that mediate the effects of different concepts. We adapt Integrated Gradients (IG; Sundararajan et al. 2017) to provide concept-aware feature attributions, by only considering gradients flowing through the hidden representation associated with a given concept. We formalize this version of IG in Appendix A.8.

In Table 4, we compare concept-aware feature attibutions for two variants of CPM$_{\text{HI}}$ (IIT and Multi-task) and the original black-box (Finetuned) model. For IIT we remove the multi-task objective $\mathcal{L}_{\text{Multi}}$ during training and for Multi-task we remove the the interchange intervention objective $\mathcal{L}_{\text{HI}}$. This helps isolate the individual effects of both losses on concept localization. All three models predict a neutral final sentiment score for the considered input, but they show vastly different feature attributions. Only IIT reliably highlights words that are semantically related to each concept. For instance, when we restrict the gradients to flow only through the intervention site of the *noise* concept, "loud" is the word highlighted the most that contributes negatively. When we consider the *service* concept, words like "friendly" and "waiter" are highlighted the most as contributing positively. These contrasts are missing for representations of the Multi-task and Finetuned models. Only the IIT training paradigm pushes the model to learn causally localized representations. For the *service* concept, we notice that the IIT model wrongfully attributes "delicious". This could be useful for debugging purposes and could be used to highlight potential failure modes of the model.

## 6   CONCLUSION

We explored the use of approximate counterfactual training data to build more robust causal explanation methods. We introduced Causal Proxy Models (CPMs), which learn to mimic both the *factual* and *counterfactual* behaviors of a black-box model $\mathcal{N}$. Using CEBaB, a benchmark for causal concept-based explanation methods, we demonstrated that both versions of our technique (CPM$_{\text{IN}}$ and CPM$_{\text{HI}}$) significantly outperform previous explanation methods without demanding the full causal graph associated with the dataset. Interestingly, we find that our GPT-3 based explanation method performs on-par with our best CPM model in some settings.

Our results suggest that CPMs can be more than just explanation methods. They achieve factual performance on par with the model they aim to explain, and they can explain their own behavior. This paves the way to using them as deployed models that both perform tasks and offer explanations. In addition, the causally localized representations of our CPM$_{\text{HI}}$ variant are very intuitive, as revealed by our concept-aware feature attribution technique. We believe that causal localization techniques could play a vital role in further model explanation efforts.

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

## A  APPENDIX

### A.1  CEBAB DATASET STATISTICS

Table 5 shows dataset statistics of CEBaB. The variants of CEBaB we consider only impact the train split. The top panel shows the number of observational samples and edits introduced in the CEBaB paper. The bottom panel shows our *paired* versions, where we create approximate counterfactual pairs. We explore two variants of approximate counterfactuals: *human*-created and *sampled* counterfactuals (Section 4.1). The *human* setting considers all pairs made possible by using *all* data. The *sampling* setting considers pairs sampled from only the *observational* data, as discussed in Section A.2.

| Dataset | # train | # dev | # test |
|---|---|---|---|
| CEBaB (*observational*) | 1,755 | 1,673 | 1,689 |
| CEBaB (*all*) | 11,728 | 1,673 | 1,689 |
| CEBaB (*paired*, *human*) | 19,684 | 3,898 | 3,958 |
| CEBaB (*paired*, *sampling*) | 74,574 | 3,898 | 3,958 |

Table 5: Dataset statistics.

## A.2 TYPES OF APPROXIMATE COUNTERFACTUAL PAIRS

Our approximate counterfactual training data comes in paired sentences of (*original sentence*, *approximate counterfactual sentence*). The approximate counterfactuals differs from their original counterparts in only one concept value. We consider approximate counterfactual pairs to be symmetric: we use both (*original sentence*, *approximate counterfactual sentence*) and (*approximate counterfactual sentence*, *original sentence*) as training pairs.

**Human-created Counterfactuals** CEBaB contains multiple counterfactual sentences for each original review. To achieve this, the dataset creators asked annotators to edit the original sentence to achieve a specified goal (e.g., 'change the evaluation of the restaurant's food to negative'). These originals and corresponding edits form our *human* pairs.

**Metadata-sampled Counterfactuals** Human-created counterfactuals are not always available. With CEBaB, we simulate a second type of approximate counterfactuals by using metadata-guided heuristics: for a given *original sentence*, we sample a counterfactual from the train set by matching concept labels while allowing only one label to be changed.

During training, we also consider *null effect pairs* in our *sampling* setup. These pairs resemble cases where our approximate counterfactual sentence is identical to the original sentence. When training our models on these pairs, we expect our models to predict the same counterfactual and factual output.

## A.3 TRAINING REGIMES

**CPM$_{IN}$** To train CPM$_{IN}$, we use the same model architecture as $\mathcal{N}$, and initialize it with the model weights using weights from $\mathcal{N}$. The maximum number of training epochs is set to 30 with a learning rate of $5e^{-5}$ and an effective batch size of 128. The learning rate linearly decays to 0 over the 30 training epochs. We employ an early stopping strategy for COS$_{ICaCE}$ over the dev set for an interval of 50 steps with early stopping patience set to 20. We set the max sequence length to 128 and the dropout rate to 0.1. We take a weighted sum of two objectives as the loss term for training CPM$_{HI}$. Specifically, we use $[w_{Mimic}, w_{IN}] = [1.0, 3.0]$. For the smoothed cross-entropy loss, we use a temperature of 2.0.

**CPM$_{HI}$** To train CPM$_{HI}$, we use the same model architecture as $\mathcal{N}$, and initialize it with the model weights using weights from $\mathcal{N}$. The maximum number of training epochs is set to 30 with a learning rate of $8e^{-5}$ and an effective batch size of 256. We use a higher learning rate of 0.001 for the LSTM model as it enables quicker convergence. The learning rate linearly decays to 0 over the 30 training epochs. We employ an early stopping strategy for COS$_{ICaCE}$ over the dev set for an interval of 10 steps with early stopping patience set to 20. We set the max sequence length to 128 and the dropout rate to 0.1. We take a weighted sum of three objectives as the loss term for training CPM$_{HI}$. Specifically, we use $[w_{Mimic}, w_{Multi}, w_{HI}] = [1.0, 1.0, 3.0]$. In Appendix A.6, we conduct a set of ablation studies to isolate the individual contributions from each objective. For the smoothed cross-entropy loss, we use a temperature of 2.0.

Our models are all implemented in PyTorch (Paszke et al., 2019) and using the HuggingFace library (Wolf et al., 2019). All of our results are aggregated over three distinct random seeds. To foster reproducibility, we will release our code repository and model artifacts to the public.

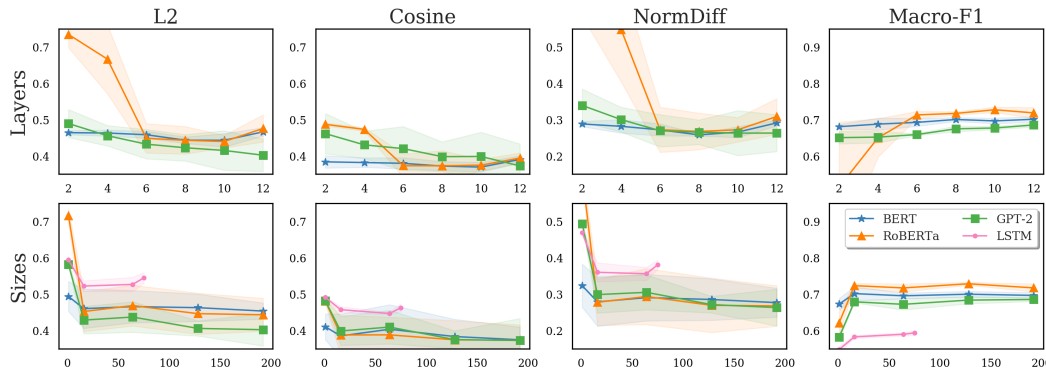

Figure 2: CEBaB scores for different intervention site locations and sizes for CPM$_{\text{HI}}$. The scores are measured in three different metrics on the test set for four different model architectures as a five-class sentiment classification task. Results averaged over three distinct seeds. Task performance as `Macro-F1` score is reported when applicable. Shaded areas outline $\pm$ SD.

## A.4    ADDITIONAL BASELINE RESULTS

Table 6 shows baselines adapted from Abraham et al. (2022), which contains the present state-of-the-art explanation methods for the CEBaB benchmark. We report the best scores across these explanation methods in Table 1. These baselines are trained without using counterfactual data. Thus, we build additional baselines that use counterfactual data as shown in Table 7. `S-Learner` is selected as the best performing models and included in Table 1 for comparisons. The equations for the additional baselines are as follows:

$$\mathcal{E}_{\mathcal{N}}^{\text{approx}}(x_{u,v}; C_i \leftarrow c') = \mathcal{N}(s^{\text{approx}}) - \mathcal{N}(x_{u,v}) \tag{10}$$

$$\mathcal{E}_{\mathcal{N}}^{\text{random}}(x_{u,v}; C_i \leftarrow c') = \mathcal{N}(s^{\text{random}}) - \mathcal{N}(x_{u,v}) \tag{11}$$

$$\mathcal{E}_{\mathcal{N}}^{\text{CaCE}}(C_i \leftarrow c') = \frac{1}{|\mathcal{D}^{C_i \leftarrow c'}|} \sum_{(x_{u,v}, \tilde{x}_{u,v}^{C_i \leftarrow c'}) \in \mathcal{D}^{C_i \leftarrow c'}} \left( \mathcal{N}\left( \tilde{x}_{u,v}^{C_i \leftarrow c'} \right) - \mathcal{N}\left( x_{u,v} \right) \right) \tag{12}$$

$$\mathcal{E}^{\text{ATE}}(C_i \leftarrow c') = \frac{1}{|\mathcal{D}^{C_i \leftarrow c'}|} \sum_{(x_{u,v}, \tilde{x}_{u,v}^{C_i \leftarrow c'}) \in \mathcal{D}^{C_i \leftarrow c'}} \left( f\left( \tilde{x}_{u,v}^{C_i \leftarrow c'} \right) - f\left( x_{u,v} \right) \right) \tag{13}$$

where $s^{\text{random}}$ is a randomly sampled training input, $s^{\text{approx}}$ is a training input sampled to match the concept-level labels of the true counterfactual under intervention $C_i \leftarrow c'$, $\mathcal{D}^{C_i \leftarrow c'}$ is the set of all approximate counterfactual training pairs that represent a $C_i \leftarrow c'$ intervention, and $f$ is a look-up function that returns the ground-truth label associated with an input.

The signatures of $\mathcal{E}^{\text{ATE}}$ and $\mathcal{E}_{\mathcal{N}}^{\text{CaCE}}$ reflect that they are independent of the specific factual input $x_{u,v}$ considered. Furthermore, $\mathcal{E}^{\text{ATE}}$ is independent of $\mathcal{N}$ given that this explainer only uses ground-truth training labels to estimate causal effects.

Additionally, we consider X-Learner, a variant of S-Learner (Künzel et al., 2019). Our X-Learner consists of three steps. First, we cluster examples into groups by their concept and predicted concept label pairs (e.g., *select all examples with food being positive*)[3]. For each group, we fit logistic regression model $\hat{\mu}_{(C_i,c)}$ to predict the factual output of black-box model $\mathcal{N}$ using concept labels for each example except for labels for $C_i$. Next, we use the models from the first step to build training sets for our individual treatment effect (ITE) estimators. To achieve this, we calculate ITE for each example as,

$$\hat{\mathcal{D}}_{u,v}^{C_i:c \leftarrow c'} = \hat{\mu}_{(C_i,c')}(\mathcal{B}(x_{u,v}^{C_i=c})') - \mathcal{N}(x_{u,v}^{C_i=c}) \tag{14}$$

---

[3]We use the finetuned concept-level sentiment analysis models $\mathcal{B}$ released by Abraham et al. (2022) for concept label prediction, which is identical to the ones used in S-Learner in Section 4.3.

| Model | Metric | Approx[†] | S-Learner[‡] | INLP[§] |
|---|---|---|---|---|
| BERT | L2 | 0.81 (.01) | 0.74 (.02) | 0.80 (.02) |
| | Cosine | 0.61 (.01) | 0.63 (.01) | 0.59 (.03) |
| | NormDiff | 0.44 (.01) | 0.54 (.02) | 0.73 (.02) |
| RoBERTa | L2 | 0.83 (.01) | 0.78 (.01) | 0.84 (.01) |
| | Cosine | 0.60 (.01) | 0.64 (.01) | 0.58 (.01) |
| | NormDiff | 0.45 (.00) | 0.59 (.01) | 0.81 (.01) |
| GPT-2 | L2 | 0.72 (.02) | 0.60 (.02) | 0.72 (.01) |
| | Cosine | 0.59 (.01) | 0.59 (.01) | 1.00 (.00) |
| | NormDiff | 0.41 (.01) | 0.40 (.01) | 0.58 (.03) |
| LSTM | L2 | 0.86 (.01) | 0.73 (.01) | 0.79 (.01) |
| | Cosine | 0.64 (.01) | 0.64 (.01) | 0.74 (.02) |
| | NormDiff | 0.50 (.01) | 0.53 (.01) | 0.60 (.01) |

Table 6: CEBaB scores measured in three different metrics on the test set for four different model architectures as a five-class sentiment classification task. Results are adapted from Abraham et al. (2022). **Lower is better**; standard deviations over 5 distinct seeds in parentheses. Results are aggregated over all aspects and all directional concept label changes. Details about these evaluation metrics can be found in Section 4. Results are based on [†]Abraham et al. (2022), [‡] Künzel et al. (2019), and [§]Ravfogel et al. (2020).

| | | no counterfactuals | sampled counterfactuals | | | | | human-created counterfactuals | | | | |
| | | | | ATE-
Explainer | CaCE-
Explainer | | | | ATE-
Explainer | CaCE-
Explainer | | |
| Model | Metric | X-Learner | Approx | Explainer | Explainer | Random | X-Learner | Approx | Explainer | Explainer | Random | X-Learner |
|---|---|---|---|---|---|---|---|---|---|---|---|---|
| BERT | L2 | 0.78 (.02) | 0.81 (.01) | 0.81 (.02) | 0.81 (.02) | 0.84 (.02) | 0.78 (.02) | 0.79 (.02) | 0.81 (.02) | 0.80 (.02) | 0.84 (.01) | 0.75 (.02) |
| | Cosine | 0.68 (.01) | 0.60 (.00) | 0.72 (.01) | 0.72 (.01) | 0.53 (.00) | 0.68 (.01) | 0.56 (.01) | 0.69 (.01) | 0.69 (.01) | 0.53 (.00) | 0.64 (.01) |
| | NormDiff | 0.53 (.03) | 0.44 (.01) | 0.62 (.02) | 0.62 (.02) | 0.55 (.02) | 0.53 (.03) | 0.43 (.01) | 0.62 (.02) | 0.64 (.02) | 0.54 (.02) | 0.54 (.03) |
| RoBERTa | L2 | 0.82 (.00) | 0.83 (.01) | 0.85 (.00) | 0.85 (.00) | 0.87 (.00) | 0.82 (.00) | 0.81 (.01) | 0.85 (.00) | 0.84 (.00) | 0.87 (.00) | 0.79 (.00) |
| | Cosine | 0.70 (.02) | 0.61 (.01) | 0.73 (.00) | 0.73 (.01) | 0.53 (.00) | 0.70 (.02) | 0.57 (.01) | 0.70 (.00) | 0.70 (.01) | 0.53 (.00) | 0.67 (.02) |
| | NormDiff | 0.57 (.00) | 0.46 (.01) | 0.67 (.00) | 0.67 (.00) | 0.58 (.00) | 0.57 (.00) | 0.44 (.01) | 0.67 (.00) | 0.68 (.00) | 0.59 (.00) | 0.58 (.00) |
| GPT-2 | L2 | 0.65 (.01) | 0.72 (.02) | 0.69 (.01) | 0.68 (.01) | 0.76 (.00) | 0.65 (.01) | 0.72 (.01) | 0.68 (.01) | 0.68 (.01) | 0.76 (.00) | 0.63 (.01) |
| | Cosine | 0.64 (.01) | 0.59 (.00) | 0.67 (.00) | 0.67 (.00) | 0.56 (.00) | 0.64 (.01) | 0.57 (.00) | 0.66 (.00) | 0.65 (.00) | 0.56 (.00) | 0.62 (.01) |
| | NormDiff | 0.41 (.00) | 0.40 (.01) | 0.48 (.01) | 0.49 (.01) | 0.47 (.00) | 0.41 (.00) | 0.40 (.00) | 0.49 (.01) | 0.50 (.01) | 0.47 (.01) | 0.42 (.01) |
| LSTM | L2 | 0.77 (.01) | 0.87 (.00) | 0.78 (.00) | 0.78 (.00) | 0.85 (.00) | 0.77 (.01) | 0.85 (.01) | 0.78 (.00) | 0.76 (.00) | 0.84 (.00) | 0.74 (.01) |
| | Cosine | 0.69 (.01) | 0.65 (.00) | 0.71 (.00) | 0.71 (.00) | 0.57 (.00) | 0.69 (.01) | 0.61 (.00) | 0.69 (.00) | 0.68 (.00) | 0.56 (.00) | 0.67 (.01) |
| | NormDiff | 0.52 (.01) | 0.50 (.00) | 0.59 (.00) | 0.59 (.00) | 0.55 (.00) | 0.52 (.01) | 0.49 (.00) | 0.59 (.00) | 0.61 (.00) | 0.55 (.00) | 0.55 (.01) |

Table 7: CEBaB scores for additional baselines we considered. CEBaB scores are measured in three different metrics on the test set for four different model architectures as a five-class sentiment classification task. **Lower is better**. Results averaged over three distinct seeds, standard deviations in parentheses. Details about these evaluation metrics can be found in Section 4.

where $\mathcal{B}(x_{u,v}^{C_i=c})'$ excludes the concept label for concept $C_i$. It measures the ITE for $x_{u,v}$ when we change the concept label of $C_i$ from $c$ to $c'$. We aggregate $\hat{\mathcal{D}}_{u,v}^{C_i:c \leftarrow c'}$ over examples based on their editing concepts and concept labels. Next, we fit a set of linear regression models as $\tau_{C_i:c \leftarrow c'}$ to predict ITE for changing the concept labels for $C_i$ given concept labels of an example except for labels for $C_i$. Lastly, we use $\tau_{C_i:c \leftarrow c'}$ to predict counterfactual output changes as,

$$\mathcal{E}_{\mathcal{N}}^{\text{X-Learner}}(x_{u,v}; C_i \leftarrow c') = p \cdot \tau_{C_i:c \leftarrow c'}(\mathcal{B}(x_{u,v}^{C_i=c})') + (1-p) \cdot \tau_{C_i:c' \leftarrow c}(\mathcal{B}(x_{u,v}^{C_i=c})') \tag{15}$$

where $p$ is the propensity score which is calculated using $\mathcal{B}$ as the probability of $C_i$ taking concept label $c'$ for an input example $x_{u,v}$ by considering two potential concept labels $c$ and $c'$.

## A.5 INTERVENTION SITE LOCATION AND SIZE

Previous work shows that neurons in different layers and groups can encode different high-level concepts (Vig et al., 2020; Koh et al., 2020). CPM$_{\text{HI}}$ pushes concept-related information to localize at the targeted intervention site (the aligned neural representations for each concept). In this section, we investigate how the location and the size of the intervention site impact CPM$_{\text{HI}}$ performance. We use the optimal location and size found in this study for other results presented in this paper.

**Location** For Transformer-based models, we vary the location of the intervention site by intervening on the "[CLS]" token embedding layer $l$. Specifically, we set $l = \{2, 4, 6, 8, 10, 12\}$. We skip this

| Model | Ablation | L2 | Cosine | NormDiff | Macro-F1 |
|---|---|---|---|---|---|
| BERT | $\mathbf{CPM_{HI}}$ | 0.45 (.02) | 0.36 (.03) | 0.27 (.04) | 0.69 (.01) |
| | $- \mathcal{L}_{\mathrm{Multi}}$ | 0.47 (.04) | 0.38 (.04) | 0.30 (.07) | 0.69 (.01) |
| | $- \mathcal{L}_{\mathrm{HI}}$ | 0.79 (.02) | 0.60 (.03) | 0.64 (.02) | 0.60 (.08) |
| | $+$ *random init* | 0.81 (.02) | 0.52 (.00) | 0.55 (.02) | 0.08 (.02) |
| | $+$ *no training* | 0.80 (.02) | 0.86 (.04) | 0.76 (.02) | 0.70 (.01) |
| RoBERTa | $\mathbf{CPM_{HI}}$ | 0.47 (.03) | 0.39 (.03) | 0.29 (.05) | 0.71 (.00) |
| | $- \mathcal{L}_{\mathrm{Multi}}$ | 0.49 (.05) | 0.41 (.05) | 0.32 (.06) | 0.70 (.00) |
| | $- \mathcal{L}_{\mathrm{HI}}$ | 0.81 (.00) | 0.53 (.02) | 0.63 (.01) | 0.39 (.06) |
| | $+$ *random init* | 0.85 (.00) | 0.51 (.00) | 0.59 (.01) | 0.06 (.00) |
| | $+$ *no training* | 0.84 (.01) | 0.93 (.05) | 0.83 (.00) | 0.70 (.00) |
| GPT-2 | $\mathbf{CPM_{HI}}$ | 0.41 (.04) | 0.39 (.05) | 0.27 (.05) | 0.68 (.00) |
| | $- \mathcal{L}_{\mathrm{Multi}}$ | 0.43 (.03) | 0.41 (.05) | 0.29 (.04) | 0.67 (.00) |
| | $- \mathcal{L}_{\mathrm{HI}}$ | 0.66 (.01) | 0.58 (.04) | 0.49 (.01) | 0.58 (.04) |
| | $+$ *random init* | 0.73 (.00) | 0.54 (.00) | 0.47 (.01) | 0.16 (.00) |
| | $+$ *no training* | 0.65 (.00) | 0.61 (.00) | 0.57 (.02) | 0.65 (.00) |
| LSTM | $\mathbf{CPM_{HI}}$ | 0.54 (.01) | 0.46 (.01) | 0.36 (.00) | 0.59 (.01) |
| | $- \mathcal{L}_{\mathrm{Multi}}$ | 0.56 (.02) | 0.47 (.02) | 0.41 (.02) | 0.59 (.01) |
| | $- \mathcal{L}_{\mathrm{HI}}$ | 0.73 (.00) | 0.64 (.02) | 0.59 (.00) | 0.59 (.01) |
| | $+$ *random init* | 0.82 (.00) | 0.55 (.00) | 0.55 (.00) | 0.13 (.04) |
| | $+$ *no training* | 0.73 (.01) | 0.74 (.00) | 0.59 (.01) | 0.60 (.01) |

Table 8: Ablation study of our $\mathrm{CPM_{HI}}$ method trained with *human* approximate counterfactual strategy. CEBaB scores measured in three different metrics on the test set for four different model architectures as a five-class sentiment classification task. **Lower is better**. Results averaged over three distinct seeds, standard deviations in parentheses.

experiment for non-Transformer-based model (i.e., LSTM) since it only contains a single sentence embedding.

As shown in the top panel of Figure 2, intervention location significantly affects $\mathrm{CPM_{HI}}$ performance. Our results show that layer 10 for BERT, layer 8 for RoBERTa, and layer 12 for GPT-2 lead to the best performance. This suggests layers have different efficacy in terms of information localization. Our results also show that intervening with deeper layers tends to provide better performance. However, for both BERT and RoBERTa, intervening on the last layer results in a slightly worse performance compared to earlier layers. This suggests that leaving Transformer blocks after the intervention site helps localized information to be processed by the neural network.

**Size** For Transformer-based models, we change the size of the intervention site $d_c$ for each concept. Specifically, we set $d_c = \{1, 16, 64, 128, 192\}$. For instance when $d_c = 1$, we use a single dimension of the "[CLS]" token embedding to represent each concept, starting from the first dimension of the vector. For our non-Transformer-based model (LSTM), we intervene on the attention-gated sentence embedding whose dimension size is set to 300. Accordingly, we set $d_c = \{1, 16, 64, 75\}$.

As shown in Figure 2, larger intervention sites lead to better performance for all Transformer-based models. For LSTM, we find that the optimal size is the second largest one instead. On the other hand, our results suggest that the performance gain from the increase of size diminishes as we increase the size for all model architectures.

## A.6 ABLATION STUDY OF $\mathrm{CPM_{HI}}$

Geiger et al. (2022) show that training with a multi-task objective helps IIT to improve generalizability. In this experiment, we aim to investigate whether the multi-task objective we added for $\mathrm{CPM_{HI}}$ plays an important role in achieving good performance. Specifically, we conduct two ablation studies: removing the multi-task objective by setting $w_{\mathrm{Multi}} = 0.0$, and removing the IIT objective by setting $w_{\mathrm{HI}} = 0.0$.

| Model | Metric | sampled counterfactuals | | | human-created counterfactuals | | |
|---|---|---|---|---|---|---|---|
| | | $\mathbf{CPM_{HI}}$ | Random Source | Probe-based Source | $\mathbf{CPM_{HI}}$ | Random Source | Probe-based Source |
| BERT | L2 | 0.60 (.01) | 0.74 (.03) | 0.61 (.01) | 0.45 (.03) | 0.70 (.03) | 0.43 (.02) |
| | Cosine | 0.45 (.00) | 0.53 (.01) | 0.45 (.00) | 0.36 (.04) | 0.59 (.04) | 0.35 (.01) |
| | NormDiff | 0.38 (.00) | 0.54 (.02) | 0.39 (.01) | 0.27 (.01) | 0.53 (.01) | 0.25 (.02) |
| RoBERTa | L2 | 0.67 (.02) | 0.79 (.01) | 0.66 (.02) | 0.47 (.03) | 0.72 (.01) | 0.44 (.01) |
| | Cosine | 0.47 (.00) | 0.52 (.01) | 0.46 (.01) | 0.39 (.03) | 0.57 (.03) | 0.37 (.01) |
| | NormDiff | 0.45 (.03) | 0.59 (.00) | 0.44 (.03) | 0.29 (.05) | 0.55 (.01) | 0.25 (.01) |
| GPT-2 | L2 | 0.51 (.01) | 0.65 (.02) | 0.51 (.02) | 0.41 (.04) | 0.58 (.03) | 0.39 (.02) |
| | Cosine | 0.46 (.00) | 0.55 (.01) | 0.46 (.01) | 0.39 (.05) | 0.56 (.02) | 0.37 (.01) |
| | NormDiff | 0.30 (.00) | 0.46 (.01) | 0.31 (.01) | 0.27 (.05) | 0.44 (.01) | 0.25 (.01) |
| LSTM | L2 | 0.64 (.02) | 0.76 (.01) | 0.65 (.02) | 0.54 (.01) | 0.69 (.03) | 0.55 (.00) |
| | Cosine | 0.50 (.01) | 0.57 (.01) | 0.50 (.01) | 0.46 (.00) | 0.58 (.01) | 0.46 (.01) |
| | NormDiff | 0.41 (.01) | 0.54 (.01) | 0.41 (.02) | 0.36 (.00) | 0.52 (.00) | 0.38 (.01) |

Table 9: Ablation study of our $CPM_{HI}$ method for different *source* input $s$ sampling strategies at inference time. CEBaB scores measured in three different metrics on the test set for four different model architectures as a five-class sentiment classification task. **Lower is better**. Results averaged over three distinct seeds, standard deviations in parentheses.

Table 8 shows our results, which demonstrate that the IIT objective is the main factor that drives $CPM_{HI}$ performance. Our results also suggest that the multi-task objective brings relatively small but consistent performance gains. Overall, our findings corroborate those of Geiger et al. (2022) and provide concrete evidence that the combination of two objectives always results in the best-performing explanation methods across all model architectures.

Additionally, we explore two baselines for $CPM_{HI}$. Firstly, we randomly initialize the weights of $CPM_{HI}$. Secondly, we take the original black-box model as our $CPM_{HI}$. Compared to the results in Table 1, these two baselines fail catastrophically, suggesting the importance of our IIT paradigm.

As mentioned in Section 3, we sample a source input $x_{u',v'}^{C_i=c'}$ from the train set as any input $x$ that has $C_i = c'$ to estimate the counterfactual output. Furthermore, we explore two additional sampling strategies. First, we create a baseline where we randomly sample a source input from the train without any concept label matching. Second, we sample a source input from the train set using the predicted concept label of our multi-task probe, instead of the true concept label from the dataset.

As shown in Table 9, the quality of our source inputs impact our performance significantly. For instance, when sampling source input at random, $CPM_{HI}$ fails catastrophically for all evaluation metrics. On the other hand, when we sampling source based on the predicted labels using the multi-task probe, $CPM_{HI}$ maintains its performance.

## A.7 GPT-3 GENERATION PROCESS

We use the 175B parameter `davinci` GPT-3 model (Brown et al., 2020) as a few-shot learner to generate approximate counterfactual data. Let $x_{u,v}$ be a review text with an original value $c$ for the mediating concept $C_i$ and an overall review sentiment $y$ (e.g., a restaurant review which is *negative* about the *service*, and felt *neutral* about their overall dining experience), and let $c'$ be the target value of $C_i$, for which we would like to create a counterfactual review (e.g., change the text to become *positive* about the mediating concept *service*). In order to use GPT-3 as an $n$-shot learner, we sample $n = 6$ approximate counterfactual pairs $(x_{u',v'}, \tilde{x}_{u',v'}^{C_i \leftarrow c'})$, where $x_{u',v'}$ shares with $x_{u,v}$ the same value $c$ for $C_i$ and the same overall sentiment, and the counterfactual review $\tilde{x}_{u',v'}^{C_i \leftarrow c'}$ has the target value $c'$ for $C_i$. We prompt the model with these pairs, and we also include the original review $x_{u,v}$. We then collect the text completed by GPT-3 as the GPT-3 counterfactual review. An example for this $n$-shot prompt and completion is in Figure 3. In addition, we also prompt GPT-3 with pairs of original reviews and metadata-sampled counterfactuals, and generate another set of GPT-3 counterfactual review for comparison. We sample $n = 4$ approximate counterfactual pairs in this case. An example of metadata-sampled counterfactual generation with GPT-3 can be seen in Figure 4.

```
Make the following restaurant reviews include POSITIVE mentions of SERVICE.

Original: I had two casual dinners at State & Lake and three lunches.  The
food was great but the service was lacking.  Everything was delicious.  The
interior is questionable, but not intrusive.

POSITIVE mentions of SERVICE: I had two casual dinners at State & Lake and
three lunches. The food and the service were always great. Everything was
delicious. The interior is questionable, but not intrusive.

Original: Food was excellent, but the service was not very attentive. Noise
level was extremely high due to close proximity of tables and poor acoustics.

POSITIVE mentions of SERVICE: Food and service was excellent.  Noise level
was extremely high due to close proximity of tables and poor acoustics.

Original: Great food, poor and very snobbish service.

POSITIVE mentions of SERVICE: Great food, very good service.

Original: My dining experince was excellent! However, the server was not nice.

POSITIVE mentions of SERVICE: My dining experince was excellent!

Original: Hae been here a few times and it is just okay - Entrees and wine
list a bit pricey for what it is, inattentive staff.

POSITIVE mentions of SERVICE: Hae been here a few times and it is just okay
- Entrees and wine list a bit pricey for what it is. Food comes out on time.

Original: Tables fairly close together, mushroom appetiser very good, pork
entree fair, chicken good. The service was terrible.

POSITIVE mentions of SERVICE: Tables fairly close together, mushroom
appetiser very good, pork entree fair, chicken good. The service was great
however.

Original: Service was very poor with the server unresponsive and misinformed
on all requests.  The food was very good with a good selection of entrees.
The ambiance was romantic with a quiet excellence.

POSITIVE mentions of SERVICE: Service was very good with the server attentive
and responsive on all requests. The food was very good with a good selection
of entrees. The ambiance was romantic with a quiet excellence.
```

Figure 3: Example GPT-3 prompt (gray) and GPT-3 completion (bold). Note that all original examples convey the same sentiment towards service ($c$ = negative) and same overall sentiment ($y$ = neutral), and that the counterfactual examples are all edited such that the sentiment towards service is the same ($c'$ = positive).

For each few-shot learning prompt, we insert an initial string of the form of "Make the following restaurant reviews include $c'$ mentions of $C_i$.", where $c'$ is expressed as one of {"POSITIVE", "NEGATIVE", "NOT" } ("NOT" corresponds to making the review be unknown regarding the concept $C_i$) and $C_i$ is one of {"AMBIANCE", "FOOD", "NOISE", "SERVICE"}. We sample using a temperature of 0.9, without any frequency or presence penalties (since we expect the counterfactual review to be similar to the original review). In preliminary experimentation, we found that capitalizing the mediating concept and target value results and inserting line breaks between examples made for better completions, although there is room for future research in this area.

```
Make the following restaurant reviews include POSITIVE mentions of SERVICE.

Original: Been here several times. Always a winner, except for the tasteless
food!

POSITIVE mentions of SERVICE: I was very disappointed in the food but we did
not wait long for each course and or waiter was very pleasant.

Original: food was decent but not great.

POSITIVE mentions of SERVICE: Lovely evening - good service and wonderful
food. Perfect for fresh fish fans

Original: The restaurant was empty when we arrived, reservation not necessary?
Wine list limited. Food was bland, presentation was very well done. I would
not eat here again.

POSITIVE mentions of SERVICE: Abby provided the best service that we've had
after probably two dozen visits. No thank you for making the risotto cake at
lunch....Two Stars!

Original: A terrible place for lunch or dinner. All the food is excellent
with top notch ingredients

POSITIVE mentions of SERVICE: Excellent Valentine's menu. Excellent service
and food. Would recommend this restaurant and will return.

Original: The food was average for the cost. My husband and I were so excited
to visit Bobby Flay's restraunt and were really disappointed. The food was
average at best.

POSITIVE mentions of SERVICE: **The service was amazing and the food was alright.**
```

Figure 4: Example `GPT-3` prompt (gray) and `GPT-3` completion (bold). Note that all original examples convey the same sentiment towards service ($c = $ unknown) and same overall sentiment ($y = $ negative), and that the counterfactual examples are all metadata-sampled such that the sentiment towards service is the same ($c' = $ positive).

We used the OpenAI API to access `GPT-3`. At the current price rate of \$0.02 per 1,000 tokens, the total cost of creating our counterfactuals (around 4,000 examples) was approximately \$50 per approximate counterfactuals creation strategy.

## A.8 INTEGRATED GRADIENTS

We adapt the Integrated Gradients (IG) method of Sundararajan et al. (2017) to qualitatively assess whether CPM$_{\text{HI}}$ learned explainable representations of mediated concepts at its intervention sites. The IG algorithm computes the average gradient from the model output to its input by incrementally interpolating from a "blank" input $x'$ (consisting only of "[PAD]" tokens) to the original input $x$. Eqn. 16 is the integrated gradients equation originally proposed in Sundararajan et al. (2017), applied to a CPM model $\mathcal{P}$ on input $x$.

$$\text{IntegratedGrads}_j(x) = (x_j - x'_j) \cdot \int_{\alpha=0}^{1} \frac{\partial \mathcal{P}(x' + \alpha \cdot (x - x'))}{\partial x_j} \partial \alpha \qquad (16)$$

Here, $\frac{\partial \mathcal{P}(x)}{\partial x_j}$ is the derivative of $\mathcal{P}$ on the $j$th dimension of $x$.

In our implementation of IG, we wish to show the per-token attribution of input $x$ on the model's final output $\mathcal{P}(x)$, *mediated by* the hidden representation of a concept in $\mathcal{P}$. That is, we'd like to ask,

| Model | Predicted | Concept | Score | Word Importance |
|---|---|---|---|---|
| Black-box | neutral | ambiance | +0.03 | [CLS] the music was too loud , and the decorations were taste ##less , but they had friendly waiter ##s and delicious pasta [SEP] |
| | | food | +0.11 | [CLS] the music was too loud , and the decorations were taste ##less , but they had friendly waiter ##s and delicious pasta [SEP] |
| | | noise | +0.04 | [CLS] the music was too loud , and the decorations were taste ##less , but they had friendly waiter ##s and delicious pasta [SEP] |
| | | service | +0.26 | [CLS] the music was too loud , and the decorations were taste ##less , but they had friendly waiter ##s and delicious pasta [SEP] |
| CPM$_{HI}$ | neutral | ambiance | −0.61 | [CLS] the music was too loud , and the decorations were taste ##less , but they had friendly waiter ##s and delicious pasta [SEP] |
| | | food | −0.88 | [CLS] the music was too loud , and the decorations were taste ##less , but they had friendly waiter ##s and delicious pasta [SEP] |
| | | noise | −1.34 | [CLS] the music was too loud , and the decorations were taste ##less , but they had friendly waiter ##s and delicious pasta [SEP] |
| | | service | +1.75 | [CLS] the music was too loud , and the decorations were taste ##less , but they had friendly waiter ##s and delicious pasta [SEP] |

Table 10: Additional visualizations of word importance scores using Integrated Gradient (IG) by restricting gradients flow through corresponding intervention site of the targeted concept. This table extends Table 4 in the main text.

"What is the effect of the word 'delicious' in the input on the model's output, when we restrict our focus only on the model's representation of the concept *food*?"

To answer this question, we compute the gradient of the model output $\mathcal{P}(x)$ with respect to the input $x$ but restrict the gradient to flow through the intervention site for a particular concept. This allows us to capture the per-token attribution of the model's final output (whether particular words contributed to a *positive*, *negative*, or *neutral* sentiment prediction), mediated by the concept that is represented by the specified intervention site. For example, in Table 4, we can see that "delicious" has a positive attribution to the output of the model when we focus on its representation of the concept *food*.

Formally, consider a trained CPM model $\mathcal{P}$, an input $x$ and mediating concept $C_i$. Let $H^{C_i}$ be the activation of $\mathcal{P}$ at the intervention site for $C_i$. We define the gradient of $\mathcal{P}(x)$ along dimension $j$, *mediated by* $C_i$, as

$$\frac{\partial \mathcal{P}(x)}{\partial x_j} \text{ mediated by } C_i = \frac{\partial \mathcal{P}(x)}{\partial H^{C_i}} \cdot \frac{\partial H^{C_i}}{\partial x_j}. \tag{17}$$

Eqn. 17 restricts the gradient to only flow through the hidden representation of the concept along which we'd like to interpret our model.

We integrate these mediated gradients over a straight path between input $x$ and baseline $x'$, analogous to Eqn. 16. We implement our IG method using `CaptumAI` library.[4] We use the default parameters for our runs with number of iterations set to 50, and we set the integral method as `gausslegendre`. We set the `multiply-by-inputs` flag to `True`. To visualize individual word importance, we conduct $z$-score normalization of attribution scores over input tokens per each concept, and then linearly scale scores between $[-1, +1]$.

Table 10 extends Table 4 in our main text with additional ablation studies on our training objectives.

## A.9  MODEL DEBIASING

Being able to accurately predict outputs for counterfactual inputs enables explanation methods to faithfully debias a model with regard to a desired concept. For instance, with CEBaB, debiasing a concept (e.g., "food") is equivalent to estimating the counterfactual output when we set the concept label for a concept to be *unknown*.

In this section, we briefly study the extent to which the CPM$_{HI}$ can function as a debiasing method. To debias a concept, we enforce the sampled source input $s$ as in Eqn. 2 to have *unknown* as its concept label for the concept to be debiased.

To show our methods can faithfully debias a targeted concept, we evaluate the correlations between the predicted overall sentiment label for sentences and the concept labels for each concept. Without any debiasing technique, we expect concept labels to be highly correlated with the overall sentiment label (e.g., if *food* is positive, it is more likely that the overall sentiment is positive). We use CPM$_{HI}$ trained for the BERT model architecture as an example, and use examples in the test set.

---

[4]https://captum.ai/

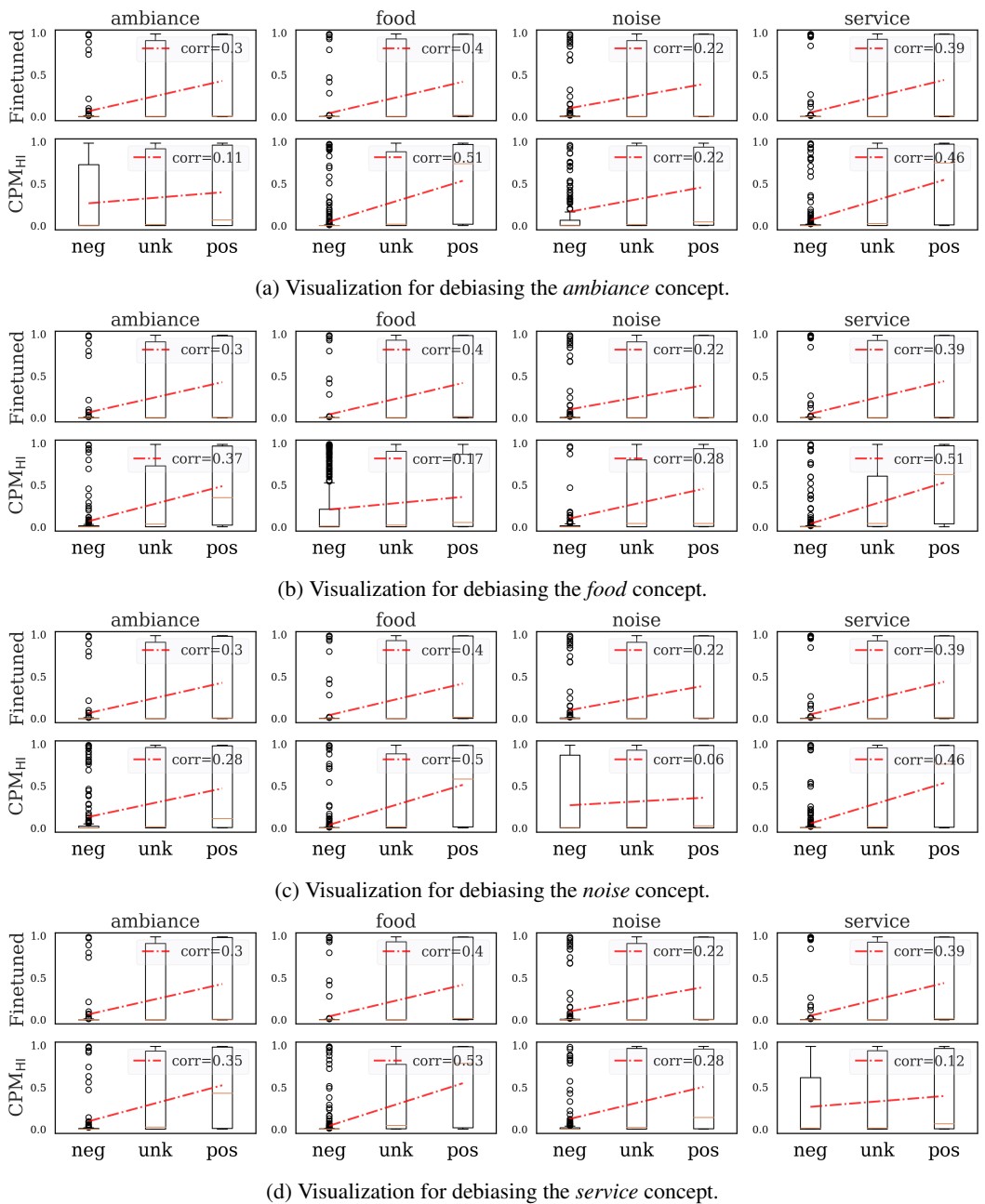

(a) Visualization for debiasing the *ambiance* concept.

(b) Visualization for debiasing the *food* concept.

(c) Visualization for debiasing the *noise* concept.

(d) Visualization for debiasing the *service* concept.

Figure 5: Debiasing visualizations for different concepts of a $CPM_{HI}$ with BERT model architecture. Individual plots are correlation plots between concept labels of a concept and the overall sentence sentiment label.

Figure 5 shows correlation plots for the black-box model as well as $CPM_{HI}$. As expected, the correlation of the *food* concept is weakened through the debiasing pipeline by 57.50%. Our results also suggest that correlations of other concepts are affected, which suggests a future research direction focused on minimizing the impact of the debiasing pipeline on irrelevant concepts. We include results for the remaining concepts in the Appendix A.9.

Figure 5a to Figure 5d show debiasing visualizations for three concepts: *ambiance*, *noise* and *service*. We use a $CPM_{HI}$ for the BERT model architecture as an example. We calculate the distributions with examples in the test set.

## A.10 LEARNING DYNAMICS

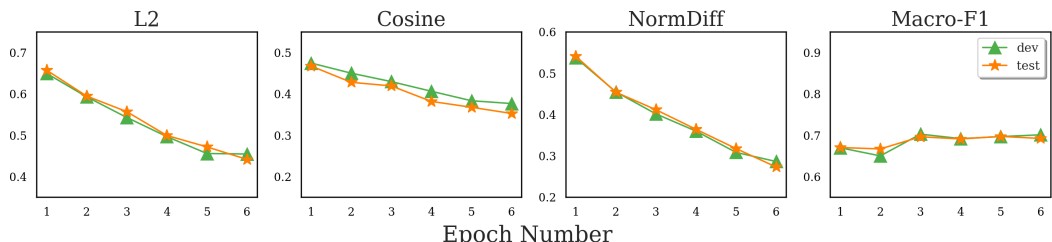

Figure 6: CEBaB scores measured in three different metrics on the dev and the test sets for a $CPM_{HI}$ with the BERT architectures for different training epochs. Task performance as Macro-F1 score is reported.

Figure 6 shows three different metrics measured on the dev and the test sets for a $CPM_{HI}$ trained for the BERT model architecture as an example. Since we use $COS_{ICaCE}$ on the dev set to early stop our training process, we find our $CPM_{HI}$ reaches a local minimum on $COS_{ICaCE}$ while $L2_{ICaCE}$ and $NormDiff_{ICaCE}$ are still trending downward. This suggests future research may need to choose desired metrics to optimize for during training, for early stopping to reach the best performing model.

| Epoch | Predicted | Concept | Score | Word Importance |
|---|---|---|---|---|
| 1 | neutral | ambiance | −0.17 | [CLS] the music was too loud , and the decorations were taste ##less , but they had friendly waiter ##s and delicious pasta [SEP] |
| | | food | +0.66 | [CLS] the music was too loud , and the decorations were taste ##less , but they had friendly waiter ##s and delicious pasta [SEP] |
| | | noise | −0.32 | [CLS] the music was too loud , and the decorations were taste ##less , but they had friendly waiter ##s and delicious pasta [SEP] |
| | | service | +0.05 | [CLS] the music was too loud , and the decorations were taste ##less , but they had friendly waiter ##s and delicious pasta [SEP] |
| 2 | neutral | ambiance | −0.25 | [CLS] the music was too loud , and the decorations were taste ##less , but they had friendly waiter ##s and delicious pasta [SEP] |
| | | food | +1.54 | [CLS] the music was too loud , and the decorations were taste ##less , but they had friendly waiter ##s and delicious pasta [SEP] |
| | | noise | −0.24 | [CLS] the music was too loud , and the decorations were taste ##less , but they had friendly waiter ##s and delicious pasta [SEP] |
| | | service | +0.02 | [CLS] the music was too loud , and the decorations were taste ##less , but they had friendly waiter ##s and delicious pasta [SEP] |
| 3 | neutral | ambiance | −0.49 | [CLS] the music was too loud , and the decorations were taste ##less , but they had friendly waiter ##s and delicious pasta [SEP] |
| | | food | +1.52 | [CLS] the music was too loud , and the decorations were taste ##less , but they had friendly waiter ##s and delicious pasta [SEP] |
| | | noise | −0.97 | [CLS] the music was too loud , and the decorations were taste ##less , but they had friendly waiter ##s and delicious pasta [SEP] |
| | | service | +0.49 | [CLS] the music was too loud , and the decorations were taste ##less , but they had friendly waiter ##s and delicious pasta [SEP] |
| 4 | neutral | ambiance | −0.69 | [CLS] the music was too loud , and the decorations were taste ##less , but they had friendly waiter ##s and delicious pasta [SEP] |
| | | food | +1.41 | [CLS] the music was too loud , and the decorations were taste ##less , but they had friendly waiter ##s and delicious pasta [SEP] |
| | | noise | −1.92 | [CLS] the music was too loud , and the decorations were taste ##less , but they had friendly waiter ##s and delicious pasta [SEP] |
| | | service | +1.14 | [CLS] the music was too loud , and the decorations were taste ##less , but they had friendly waiter ##s and delicious pasta [SEP] |
| 5 | neutral | ambiance | −0.77 | [CLS] the music was too loud , and the decorations were taste ##less , but they had friendly waiter ##s and delicious pasta [SEP] |
| | | food | +1.25 | [CLS] the music was too loud , and the decorations were taste ##less , but they had friendly waiter ##s and delicious pasta [SEP] |
| | | noise | −1.63 | [CLS] the music was too loud , and the decorations were taste ##less , but they had friendly waiter ##s and delicious pasta [SEP] |
| | | service | +1.28 | [CLS] the music was too loud , and the decorations were taste ##less , but they had friendly waiter ##s and delicious pasta [SEP] |
| 6 | neutral | ambiance | −0.66 | [CLS] the music was too loud , and the decorations were taste ##less , but they had friendly waiter ##s and delicious pasta [SEP] |
| | | food | +0.62 | [CLS] the music was too loud , and the decorations were taste ##less , but they had friendly waiter ##s and delicious pasta [SEP] |
| | | noise | −0.90 | [CLS] the music was too loud , and the decorations were taste ##less , but they had friendly waiter ##s and delicious pasta [SEP] |
| | | service | +2.14 | [CLS] the music was too loud , and the decorations were taste ##less , but they had friendly waiter ##s and delicious pasta [SEP] |
| $CPM_{HI}$ | neutral | ambiance | −0.61 | [CLS] the music was too loud , and the decorations were taste ##less , but they had friendly waiter ##s and delicious pasta [SEP] |
| | | food | −0.88 | [CLS] the music was too loud , and the decorations were taste ##less , but they had friendly waiter ##s and delicious pasta [SEP] |
| | | noise | −1.34 | [CLS] the music was too loud , and the decorations were taste ##less , but they had friendly waiter ##s and delicious pasta [SEP] |
| | | service | +1.75 | [CLS] the music was too loud , and the decorations were taste ##less , but they had friendly waiter ##s and delicious pasta [SEP] |

Table 11: Visualizations of word importance scores using Integrated Gradient (IG), using the same methods as in Table 4 and Table 10.

Table 11 visualizations of word importance scores using our version of Integrated Gradient (IG). Different from Table 4 and Table 10, which show the visualizations of our optimized model, we show a per-epoch result for for $CPM_{HI}$, followed with our best model appended at the end. Our results suggest that early checkpoints in the training process focus at drastically different input words comparing to later checkpoints, though all models predict *neutral* for this given sentence. In addition, gradient aggregations over input words are rather stable towards the end the training. More importantly, $CPM_{HI}$ learns how to highlight words that are semantically related to each concept gradually. For instance, we can see a clear trend of emphasising the word "decorations" for the *ambiance* concept throughout the training process. This suggests that our training procedure induces causally motivated gradients over input words gradually through the training process.

