# OpenReview forum: "Causal Proxy Models For Concept-Based Model Explanations"
_ICLR.cc/2023/Conference — Submitted to ICLR 2023_

### Official Review · Reviewer_aAEJ · 2022-10-15

**Confidence:** 3
**Correctness:** 3
**Technical Novelty And Significance:** 3
**Empirical Novelty And Significance:** 2
**Recommendation:** 5

**Clarity, Quality, Novelty And Reproducibility:**

On the whole I think the paper was fairly clear -- however, as someone slightly outside this area I did have trouble understanding the motivations and methods on the first pass. One thing that I think would help would be more linguistic examples illustrating the kinds of inputs and counterfactual edits are being used, to anchor the descriptions in the specific phenomena being targeted.

I think the quality and reproducibility of the work are decent, and I don't know any other work similar enough to indicate an issue in novelty. My main issue, as outlined above, is that the methods feel very narrowly applicable, and I'm not seeing a clear indication that they produce the kinds of interpretable insights that I would expect from an explainability method.

**Strength And Weaknesses:**

Strengths: The paper presents an interesting set of methods, which are explained in the paper reasonably clearly, and which show state-of-the-art performance on the CEBaB dataset. The authors include a couple of follow-up analyses that help to flesh out the overall picture of the methods' outcomes and potential impact.

Weaknesses: My main concern with the paper is that the methods feel very narrowly focused on improving on the CEBaB dataset, which seems like an interesting task, but which casts explanation in a very particular way. So there are two subparts to this concern. First, the way the proxy models are trained feels very specifically targeted toward improving on CEBaB -- aka, leveraging the fact that the task is focused on predicting how models will behave on inputs edited for particular sentiment features -- so although the methods are interesting, I'm not sure how surprising I find it that they were able to improve over other methods, and I'm also skeptical about how generalizable this method would be for explanation on other types of domains/tasks/explainability settings.

The second subpart of the concern relates to the last point: while the authors are targeting an existing benchmark that has been introduced for explainability, and they show improvement on that benchmark, it's just not clear to me from the paper in what way their method is actually resulting in interpretable/usable insights for explaining the behaviors of the models being approximated. What they show is that their methods allow them to use feature tokens or representational interventions to predict how target model predictions will change if an input has been changed along a particular sentiment feature dimension. But it's not clear to be how this help us to understand what strategies the models are using, what features are influential for prediction, what potential heuristics might cause untrustworthy behavior, etc -- the types of things that I expect from an explanation method. I'm happy to believe that there is more insight here than was made clear, but at the moment I don't feel the paper makes a strong argument for how it is providing insights into the target models.

**Summary Of The Paper:**

This paper presents Causal Proxy Models, which are trained to mimic outputs of a given model for original inputs as well as for alternative (counterfactual) versions of those inputs that differ based on modification of a particular (sentiment) feature. They use data from the CEBaB dataset for creating human-annotated and sampled input pairs, and train their models on these data. The models involve either a special token indicating change of a particular feature, or representation intervention on representations trained to store causally-relevant information for a given feature. The authors report that their methods achieve state-of-the-art performance on the CEBaB dataset. They also show that the models can explain their own behavior, have performance on par with the original models, and that the representations trained to capture certain features do show reasonable feature attributions.

**Summary Of The Review:**

The proposed methods are interesting, and show improvement on the specific benchmark being used. However, I have concerns about the narrowness in applicability of the method (not clear it's useful beyond that specific benchmark) and I'm not clear how much insight it provides as a contribution to explainability.

---

> ### Author Response · Authors · 2022-11-09
> **Official Response to Reviewer aAEJ**
>
> We greatly appreciate the reviewer for their *in-depth feedback on our paper*, which leads us to be able to *better situate our work in the landscape of different explanation methods*. Below we provide a point-to-point response, which adds to our above response to the shared reviewer themes.
>
> > Q1: *``My main concern with the paper is that the methods feel very narrowly focused on improving on the CEBaB dataset.’’*
>
> **A1**: **To our best knowledge, CEBaB is the first naturalistic dataset in evaluating causal concept-based model explanation methods in NLP. CPMs are the best performing model on CEBaB.** We do think this provides useful general insights into the design of explainer methods. In particular, one can obtain very strong CPM explainers using only meta-data based approximate counterfactuals of a sort that are abundant in natural datasets, since one really just needs intermediate ratings of concepts and a final rating. Most consumer websites supply such data, for example. We envision CPMs attaining strong explanation performance on many NLP tasks, either through sampling meta-data based approximate counterfactuals or human counterfactual creation efforts.
>
> > Q2: *``I'm not sure how surprising I find it that they were able to improve over other methods, and I'm also skeptical about how generalizable this method would be for explanation on other types of domains/tasks/explainability settings.’’*
>
> **A2**: We suppose it is inevitable that using benchmarks as measurement tools will shape what models people choose to explore. That said, the original CEBaB paper evaluates a lot of existing methods. They are used *``off the shelf’’* there, and we have no reason to believe that they are being mis-used. Indeed, CEBaB seems like a very suitable application scenario for them. Our CPMs beat all of these methods. **One might be concerned that CEBaB is a fairly easy task, but many methods perform only slightly better than chance on it.**
>
> More generally, CEBaB seems like a useful measurement tool for any scenario in which one has intermediate concepts that mediate an overall outcome. This covers many real-world scenarios.
>
> > Q3: *``it's just not clear to me from the paper in what way their method is actually resulting in interpretable/usable insights for explaining the behaviors of the models being approximated … it's not clear to be how this help us to understand what strategies the models are using, what features are influential for prediction, what potential heuristics might cause untrustworthy behavior, etc’’*
>
> **A3**: These are very interesting questions for us. The CEBaB protocols measure how well an explainer can estimate counterfactual model behavior. Access to such an explainer is valuable for many real-world applications. By cheaply estimating counterfactual model behavior at test-time (without access to counterfactual inputs at test-time), users can quickly get an elevated understanding of the model behavior. While we show impressive explanation behavior on the CEBaB benchmark, we explore a different notion of interpretability afforded by our methods. We conduct integrated gradient analyses of the $\textbf{CPM}_\textbf{HI}$ models and show that these CPMs are localizing concept-level information strongly. This last result is more qualitative at present, but it aligns with the benchmark results.
>
> We do feel that IG is trustworthy when it comes to making causal inferences about the role of specific neurons in model behavior. IG has a natural causal interpretation.

---

### Official Review · Reviewer_MZ4t · 2022-10-23

**Confidence:** 4
**Clarity, Quality, Novelty And Reproducibility:** Please check what I have discussed ab…
**Correctness:** 3
**Technical Novelty And Significance:** 2
**Empirical Novelty And Significance:** 2
**Recommendation:** 3

**Strength And Weaknesses:**

### Strengths
1) I think the main strength of the paper is that it is well-written and easy to follow.
2) Learning a model that enables the explanation of a pretrained black-box model has a major practical impact. However, the proposed method is limited to a particular scenario in which counterfactual data is available.

### Weaknesses
I) About the motivation and idea:
1) In the Abstract and Introduction, the authors argue that *“the fundamental problem of causal inference is that we hardly observe the counterfactual inputs”* but the proposed method, by design, assumes (approximate) counterfactuals are available. This limits the practical application of the proposed method to certain datasets (e.g., the CEBaB dataset in the paper) and cases.
2) I do not see any discussion about the clear drawbacks of existing methods for explanation in comparison with the proposed method.

II) About the proposed method:
1) In my opinion, the proposed method is not very novel. Given counterfactual data $\tilde{x}^{C_i \leftarrow c'}\_{u,v}$,  it is quite straightforward to think of matching $\tilde{x}^{C_i \leftarrow c’}\_{u,v}$ with $x_{u,v}; t\_{C_i \leftarrow c’}$ for convenient intervention.
2) A limitation of CPM is that it does not account for the stochasticity of $\tilde{x}^{C_i \leftarrow c’}\_{u,v}$. $x_{u,v}; t\_{C_i \leftarrow c’}$ only yield one value but the corresponding counterfactual text $\tilde{x}^{C_i \leftarrow c’}\_{u,v}$ can be abundant.

III) About the presentation:
1) The overall writing is good but the descriptions of many (mathematical) terms in the paper are not clear, causing difficulty in understanding the method. I would like to discuss some of them below:
- I don’t really understand what is the *“description of a concept intervention $C_i \leftarrow c’$”*. Does it mean $C_i$ will take one of three categorical values {negative, unknown, positive} or a text that describes the concept $C_i$? Can we just call $C_i\leftarrow c’$ a concept intervention for simplicity?
- What is the token $t\_{C_i \leftarrow c’}$ used in the paper? Would the authors please provide some concrete examples of this variable as I could not find such thing in the paper? Is it possible to just write $t\_{c’}$ instead of $t\_{C_i \leftarrow c’}$?
- The objective $\mathcal{L}\_{\text{mimic}}$ seems to be shared between $\text{CPM}\_\text{IN}$ and $\text{CPM}_\text{HI}$ and should be put above the paragraph that describe $\text{CPM}\_\text{IN}$. The authors should also write an overall loss, e.g., $\mathcal{L} = \lambda_1 \mathcal{L}\_{\text{mimic}} + \lambda\_2 \mathcal{L}\_{\text{counterfact}}$ where $\mathcal{L}\_{\text{counterfact}}$ can be either $\mathcal{L}\_{\text{IN}}$ or $\mathcal{L}\_{\text{HI}}$
2) The Structural Causal Model (SCM) described in Figure. 1a does not really make sense to me. I cannot figure out what are treatment, covariate, and outcome from this figure. From what shown in the figure, I assume $C_1$, …, $C_k$ are different treatments, X is an outcome, and no covariate. But it turns out X is actually a covariate, and the outcome is a 5-star sentiment score Y which does not appear in the causal graph. I suggest the authors to redraw the SCM to make it correct, adding Y and removing U, V as they do not have any contribution to the model.
3) For the S-learner baseline described in Section 4.3, I don’t really understand how the authors actually perform intervention with the output of the BERT model $\mathcal{B}$. In other words, how is  $\mathcal{B}(x_{u, v})_{C_i \leftarrow c’}$ implemented?

IV) About experiments:
1) Please provide references for methods used in $\text{BEST}_{\text{CEBaB}}$.
2) S-learner considered in the paper seems not a strong baseline since the intervention is only done for the output of the BERT model on the FACTUAL text $x_{u, v}$ which means “late fusion” between covariate $X$ and treatment $C_i$. Meanwhile, CPM fuses $X$ and $C_i$ early, which allows the method to model the interaction between $C_i$ and $X$ better.
3) The authors should compare their method with other causal inference methods such as X-learner [1], TARNet [2], and Balancing Linear Regression [3] in a fairer setting. The authors should also compare their method with other methods for explanation discussed in the paper.
4) In this paper, the authors consider $\mathcal{N}$ as a black-box model but in the experiment, they initialize $\mathcal{P}$ from the weights of a pretrained $\mathcal{N}$. This means $\mathcal{N}$ is no longer black-box anymore. I suggest the authors to do experiments with $\mathcal{P}$ different from $\mathcal{N}$ (e.g., $\mathcal{N}$ is BERT and $\mathcal{P}$ is GPT-2 or LSTM). This will make the method more convincing.

[1] Metalearners for estimating heterogeneous treatment effects using machine learning, Kunzel et al., PNAS-2019

[2] Estimating individual treatment effect: generalization bounds and algorithms, Shalit et al., ICML-2017

[3] Learning Representations for Counterfactual Inference, Johansson et al., ICML-2016

**Summary Of The Paper:**

This paper proposes a method called Causal Proxy Model (CPM) for explaining a black-box model $\mathcal{N}$ that makes use of available counterfactual training data.

**Summary Of The Review:**

This paper is a well-written paper. However, its problem is not very significant and limited to certain cases, and its proposed solution is quite straightforward. Thus, I think the paper is below the acceptance bar of ICLR.

---

> ### Author Response · Authors · 2022-11-09
> **Official Response to Reviewer MZ4t (Part 3/3)**
>
> ### **IV) About experiments**:
>
> > Q11: *``Please provide references for methods used in $\text{BEST}_{\text{CEBaB}}$.’’*
>
> **A11**: **We show that CPMs beat all 7 methods evaluated in CEBaB by wide margins** including *Approx* [Abraham et. al., 2022], *CONEXP* [Goyal et. al., 2020], *S-Learner* [Kunzel et. al., 2019], *TCAV* [Kim et. al., 2018], *ConceptSHAP* [Yeh et. al., 2020], *INLP* [Ravfogel et. al., 2020] and *CausalLM* [Feder et. al., 2021], and CPMs even beat our improved version of the best of those methods (e.g., *S-Learner*).
>
> > Q12: *``S-learner considered in the paper seems not a strong baseline since the intervention is only done for the output of the BERT model on the FACTUAL text … Meanwhile, CPM fuses  $X$ and $C_{i}$ early, which allows the method to model the interaction ....’’*
>
> **A12**: This is an incisive characterization of CPMs, we thank the reviewer for this remark. *S-Learner* [Kunzel et. al., 2019] is the best method in the CEBaB paper. We picked it to show SoTA results due to space constraints. **Our baselines, in fact, cover a much wider range of methods, including early fusion (e.g, *GPT-3*)**, late fusion, probing based (e.g., *CausalLM* [Feder et. al., 2021]), *INLP* [Ravfogel et. al., 2020], *TCAV* [Kim et. al., 2018], *CONEXP* [Goyal et. al., 2020], etc.. In fact, taking baselines included in the CEBaB paper [Abraham et. al., 2022], we evaluate our methods against more than 7 methods, and still establish the SoTA results on CEBaB. The competition is not even close (see Table 10 in the Abraham et al, paper for a full accounting). While *S-Learner* does not allow for early fusion of the intervention information, our *GPT-3* baseline method does. Nevertheless, our CPMs beat *GPT-3* behavior, indicating that the strong behavior is not uniquely due to early fusion.
>
> > Q13: *``The authors should compare their method with other causal inference methods such as X-learner [1], TARNet [2], and Balancing Linear Regression [3] in a fairer setting.’’*
>
> **A13**: Useful suggestions! We are working on building an enhanced *X-learner* [1] explainer following our *S-learner* implementation, and will report numbers, if not during the author response then for the next version of the paper. We are also working on adapting *Balancing Linear Regression* [3] for providing causal model explanations.
>
> This adaptation process is informative for us. These methods have not traditionally been included in efforts to develop model explanation methods [Feder et. al., 2021]. X-learner is designed for estimating real-world effects of data using a model. Our goal is model explanation. In technical terms, one can move between these two domains, since real-world effects are based in ATE values whereas model explanations are based in CaCE values, which are superficially similar quantities. However, the goals of model explanation are very different. We want to explain models even when (maybe especially when) they are not good estimators of real-world effects!
>
> > Q14: *``The authors should also compare their method with other methods for explanation discussed in the paper.’’*
>
> **A14**: Since we follow all the same protocols as the original CEBaB paper, we are comparing against all those models using the CEBaB benchmark. As we noted above, this means we are comparing against more than 7 methods right now. We will update our text to reflect this more clearly.
>
> > Q15: *``In this paper, the authors consider $\mathcal{N}$ as a black-box model but in the experiment, they initialize $\mathcal{P}$ from the weights of a pretrained. This means $\mathcal{N}$ is no longer black-box anymore. I suggest the authors to do experiments with $\mathcal{P}$ different from $\mathcal{N}$.’’*
>
> **A15**: Yes, our CPM models are initialized with the weights of $\mathcal{N}$. However, this is no strict requirement of our method. Many of the methods benchmarked on CEBaB assume a similar access to the factual model when formulating explanations. For all these methods, the model weights for $\mathcal{N}$ remain frozen during training. We think that additional experiments such as randomly initializing CPM models, or initializing CPM models with pretrained language model weights, or compressing the model parameters of CPM are very interesting to pursue as future directions.

---

> > ### Author Response · Authors · 2022-11-15
> > **Additional Response with X-Learner Results and Revised Paper**
> >
> > Thanks again for your suggestions on additional baselines. **We implemented X-Learner, and we provide details in Appendix A.4. Performance results of X-Learner in various condition are included in Table 7.** X-Learner achieves performance similar to S-Learner with metadata-sampled counterfactuals, and performs less competitive with human-created counterfactuals. As a result, our CPMs outperforms X-Learner by large margins. We also include the X-Learner results here for your reference:
> >
> > | Model   | Metric   | S-Learner (sampled) | X-Learner (sampled) |   | S-Learner (human-created) | X-Learner (human-created) |
> > |---------|----------|---------------------|---------------------|---|---------------------------|---------------------------|
> > | BERT    | L2       | **0.74**            | 0.78                |   | **0.73**                  | 0.75                      |
> > | BERT    | COS      | **0.63**            | 0.68                |   | **0.60**                  | 0.64                      |
> > | BERT    | NormDiff | **0.53**            | **0.53**            |   | **0.52**                  | 0.54                      |
> > | RoBERTa | L2       | **0.78**            | 0.82                |   | **0.77**                  | 0.79                      |
> > | RoBERTa | COS      | **0.65**            | 0.70                |   | **0.63**                  | 0.67                      |
> > | RoBERTa | NormDiff | 0.58                | **0.57**            |   | **0.56**                  | 0.58                      |
> > | GPT-2   | L2       | **0.61**            | 0.65                |   | **0.61**                  | 0.63                      |
> > | GPT-2   | COS      | **0.59**            | 0.64                |   | **0.59**                  | 0.62                      |
> > | GPT-2   | NormDiff | **0.41**            | **0.41**            |   | **0.40**                  | 0.42                      |
> > | LSTM    | L2       | **0.73**            | 0.77                |   | **0.72**                  | 0.74                      |
> > | LSTM    | COS      | **0.64**            | 0.69                |   | **0.63**                  | 0.67                      |
> > | LSTM    | NormDiff | 0.53                | **0.52**            |   | **0.54**                  | 0.55                      |

---

> ### Author Response · Authors · 2022-11-09
> **Official Response to Reviewer MZ4t (Part 2/3)**
>
> > Q6: *``I don’t really understand what is the description of a concept intervention $C_{i} \leftarrow c’$. Does it mean $C_{i}$  will take one of three categorical values {negative, unknown, positive} or a text that describes the concept $C_{i}$? Can we just call $C_{i} \leftarrow c’$ a concept intervention for simplicity?’’*
>
> **A6**: We would like to distinguish between a **``description of a concept intervention’’** and a **``concept intervention‘’**. A **``concept intervention”** can be **``described”** in many ways, e.g., as natural language (*``change the food to negative’’*) or as a special learned representation. In our case, $t_{C_{i} \leftarrow c’}$ is a special learnable token appended to the input, which describes the intervention. These tokens are newly initialized for this purpose and the CPM learns to interpret them accordingly only via our counterfactual training procedure. For GPT-3, we use $C_{i} \leftarrow c’$’ to denote a set of prompts in natural languages describing the intervention, see Appendix A.7.
>
> > Q7: *``What is the token $t_{C_{i} \leftarrow c’}$ used in the paper? Would the authors please provide some concrete examples of this variable as I could not find such thing in the paper? Is it possible to just write $t_{c’}$ instead of $t_{C_{i} \leftarrow c’}$?’’*
>
> **A7**: In our setting, $t_{C_{i} \leftarrow c’}$ is simply a learned embedding appended to the input as a post-fix that tells the model our intervention type (e.g., change food to negative). This token is randomly initialized and only gains meaning during our counterfactual training procedure. It could also be a set of tokens describing this in natural language appended to the input (e.g., ``the food is good but not the service $\texttt{[SEP]}$ change food to negative’’), but we do not explore this. We can not replace $t_{C_{i} \leftarrow c’}$ with $t_{c’}$, because it is crucial our token carries information about the concept that is intervened upon (e.g. food) and the value it takes after intervention (e.g. negative).
>
> > Q8: *``The objective $\mathcal{L}_{\text{mimic}}$ seems to be shared between $\textbf{CPM}_\textbf{IN}$ and $\textbf{CPM}_\textbf{HI}$ and should be put above the paragraph that describe $\textbf{CPM}_\textbf{IN}$. The authors should also write an overall loss.’’*
>
> **A8**: Thanks! We will add this to our next revision. Thank you for this suggestion.
>
> > Q9: *``The Structural Causal Model (SCM) described in **Figure 1a** does not really make sense to me. I cannot figure out what are treatment, covariate, and outcome from this figure.’’*
>
> **A9**: Thanks for this! We will clarify the role of Figure 1a in our next version. Our intention was for Figure 1a to describe the generating process for utterances in the benchmark. In this, we followed the original CEBaB paper. It seems to us that you are correct that Figure 1a should be expanded in some way to clarify the role of the final sentiment rating predicted by the model. We will add two new variables to these graphs that take on the value of logit vectors produced by the neural model.
>
> We are nervous about removing $\textit{U}$ and $\textit{V}$. **They are crucial in illustrating how we sample our approximated counterfactuals during training.** This subtle use of $\textit{U}$ and $\textit{V}$ is highlighted in **Figure 1e**.
>
> > Q10: *``For the S-learner baseline described in **Section 4.3**, I don’t really understand how the authors actually perform intervention with the output of the BERT model’’*
>
> **A10**: Yes, this is very compressed in the current paper. We will clarify how we use model $\mathcal{B}$. Model $\mathcal{B}$ is completely different and separate from N. It is a new model trained to predict intermediate concept labels, which is needed as input for *S-Learner* [Kunzel et. al., 2019]. We use a BERT model for $\mathcal{B}$, but it can be any model architecture. These predicted concept labels are fed into *S-Learner*, and *S-Learner* is trained to predict the model N output. Thus, *S-Learner* does not rely on concept labels being accessible during inference time, they can be predicted. Because the output of this BERT model is a ternary label indicating the value for each predicted concept, we can easily intervene on these outputs before feeding them to *S-Learner* by manually manipulating the ternary labels. For example, if BERT predicts that the food is negative, we can manually set it to positive while keeping all other predicted concepts by BERT constant, before feeding the predictions to *S-Learner*. This constitutes an intervention in S-Learner’s input space.

---

> ### Author Response · Authors · 2022-11-09
> **Official Response to Reviewer MZ4t (Part 1/3)**
>
> We would like to thank the reviewer for their *impressive in-depth feedback* on our paper, which leads us to be able to *better explain advantages of the method we are proposing*. Below we provide a point-to-point response organized by sections, which adds to our above response to the shared reviewer themes.
>
> ### **I) About the motivation and idea**:
>
> > Q1: *``In the Abstract and Introduction, the authors argue that the fundamental problem of causal inference is that we hardly observe the counterfactual inputs’’ but the proposed method, by design, assumes (approximate) counterfactuals are available.’’*
>
> **A1**: Yes, absolutely! But **they can be very, very approximate and still yield useful explainers!** We push the limits here: our meta-data sampled counterfactual approach depends only on the sort of counterfactuals one can obtain in abundance on the web. For example, any consumer site where customers offer specific ratings and overall ratings will support our approach. This shows that current explainers should be making much more use of (very) approximate counterfactuals. Additionally, our methods only require train-time access to approximated counterfactuals, providing a user with faithful test-time counterfactual explanations without test-time counterfactual inputs. Our methods can increase the explanation performance on many relevant tasks, if steps are first undertaken to approximate some counterfactual training data via crowdsourcing or sampling.
>
> **CEBaB is special here only insofar as any benchmark tool: it offers what we hope is a reliable measurement tool.**
>
> > Q2: *``I do not see any discussion about the clear drawbacks of existing methods for explanation in comparison with the proposed method.’’*
>
> **A2**: Due to space constraints, we do partly have to use the benchmark to make this case. This evidence shows that CPMs achieve better results than any of the methods evaluated in the original CEBaB paper, and that they are better than our strengthened version of the best of those methods (e.g., *S-Learner* [Kunzel et. al., 2019]).
>
> ### **II) About the proposed method**:
>
> > Q3: *``In my opinion, the proposed method is not very novel. Given counterfactual data* *$\tilde{x}_{u, v}^{C_i \leftarrow c’}$* *, it is quite straightforward to think of matching* *$\tilde{x}_{u, v}^{C_i \leftarrow c’}$* *with* *$\tilde{x}_{u, v}$*;*$t_{C_i \leftarrow c’}$* *for convenient intervention.’’*
>
> **A3**: This is a description of $\textbf{CPM}_\textbf{IN}$. $\textbf{CPM}_\textbf{IN}$ is our vanilla setup where the intervention happens at the input level. However,  **$\textbf{CPM}_\textbf{IN}$ is not the main focus for our paper, $\textbf{CPM}_\textbf{HI}$ is our focus**, as it not only provides great concept-based explanations compared to other methods, but it also provides feature attributions by its own learned representations. $\textbf{CPM}_\textbf{HI}$ operates differently. It needs an intervention in the hidden representation space where convenient intervention does not exist. Training models under these interventions is not trivial and underexplored in the machine learning literature.
>
> > Q4: *``A limitation of CPM is that it does not account for the stochasticity of* *$\tilde{x}_{u, v}^{C_i \leftarrow c’}$*. *$\tilde{x}_{u, v}$* *;* *$t_{C_i \leftarrow c’}$* *only yield one value but the corresponding counterfactual text* *$\tilde{x}_{u, v}^{C_i \leftarrow c’}$* *can be abundant.’’*
>
> **A4**: Agreed! $\textbf{CPM}_\textbf{IN}$ does not account for stochasticity because it always appends the same token for a specific intervention. However, $\textbf{CPM}_\textbf{HI}$ incorporates hidden representations coming from a source input. The specific source input that is chosen is variable, and thus introduces stochasticity in the process. Understanding how this stochasticity relates to the distribution of counterfactual model outputs that we can observe is an interesting question. We will continue to think about this in our future work.
>
> ### **III) About the presentation**:
>
> > Q5: *``The overall writing is good but the descriptions of many (mathematical) terms in the paper are not clear, causing difficulty in understanding the method.’’*
>
> **A5**: Thank you for this detailed feedback! This will help us to make the paper more accessible in our next revision!

---

### Official Review · Reviewer_LSZt · 2022-10-24

**Confidence:** 2
**Correctness:** 4
**Technical Novelty And Significance:** 3
**Empirical Novelty And Significance:** 3
**Recommendation:** 6

**Clarity, Quality, Novelty And Reproducibility:**

The paper is well-written and the idea is clearly stated. However, I am not an expert in NLP area. Thus, I don't think it is fair to evaluate its novelty.

**Strength And Weaknesses:**

Strength:
1) The paper proposes to use CPM to approximate the factual and counterfactual performance of a black-box model, which is an interesting idea.
2) The learned CPM can perform comparably to the black-box model in terms of factual performance.
3) CPM_HI generates a conceptual level of understanding of the black-box model.
4) The paper provides the insight to directly use CPM as the deployed model and explainer.

Weaknesses:
1) The causal graph in Figure 1a is particularly for CEBaB data set. Does the method work without knowing the causal graph? My understanding is that one has to know the causal graph so as to use CPM. This can be its limitation.

**Summary Of The Paper:**

The paper proposes a robust explanation method for NLP tasks using Causal Proxy Model (CPM). Given a black-box model to be explained, the CPM tries to simulate both the factual and counterfactual performance of that model. With CPM, one can have 1) an explanation of the black-box model, 2) comparable factual performance and 3) learn concept-level representation in the hidden code. The model is evaluated on the CEBaB benchmark.

**Summary Of The Review:**

Overall, the paper is well-written and the story follows naturally. The proposed method seems interesting and novel to me. In particular, the paper builds the insight that CPM can be directly used in the replacement of a black-box model. However, one potential limitation is that it builds on a known causal graph, which is usually unavailable in the world.

---

> ### Author Response · Authors · 2022-11-09
> **Official Response to Reviewer LSZt**
>
> We *greatly appreciate* the reviewer for their feedback on our paper, which leads us to be able to *better explain how our methods depend on causal graph*. Below we provide a point-to-point response, which adds to our above response to the shared reviewer themes.
>
> > Q1: *``The causal graph in **Figure 1a** is particularly for CEBaB dataset. Does the method work without knowing the causal graph?’’*
>
> **A1**: Excellent questions (and the summary provided really gets at the essence of what we are going for with CPMs)! We will clarify this in the paper. In brief:
> - **Our method does not require the full causal graph to be known.** A partial causal graph is fine. Indeed, the full causal graph including the counterfactual examples produced by it may never be known. We allude to this in the intro and abstract and will amplify this in the next version.
> - **Explainers are useful only to the extent that they can work with partial causal graphs!** Our technique can increase explanation performance on many tasks where a full causal graph is not known, but where it is possible to make enough assumptions about the graph to either sample some approximated counterfactuals or task humans with creating some counterfactual data.

---

### Official Review · Reviewer_9V3t · 2022-10-25

**Confidence:** 4
**Correctness:** 2
**Technical Novelty And Significance:** 2
**Empirical Novelty And Significance:** 2
**Recommendation:** 6

**Clarity, Quality, Novelty And Reproducibility:**

The methodology and experiments proposed in this paper, are not well geared toward X-AI research. Nonetheless, the authors will release the code. They articulated the hyperparameters and training strategies clearly in the paper.


**Strength And Weaknesses:**

# Strength
1. Having a model that is counterfactually consistent is a reasonable goal when the model us to interact with humans.
 2. the experiments demonstrated that the proposed CPM performs better than the baselines for various architectures.
 3. Their experiments also showcased that the CPM can be consistent with the BlackBox and can even replace the BlackBox for estimating factual and counterfactual outcomes.

# Weakness
1. My biggest concern with this paper is motivation and how it is presented. The overall aim is to have counterfactual consistency, meaning the output should change reasonably by changing the counterfactual concept. It is not clear to this reviewer why they call their method an explanation. The so-called CPM is a fine-tuned version of the original model (with the same arch.), hence using the integrated gradient to explain CPM is not the same as explaining the original model.  Overall, I do not agree with the authors that this is an explanation method. They did not compare with some existing explanation strategies like TCAV, demonstrating how these counterfactual explanations serve a better purpose. They should rename the paper if the original aim is to predict good counterfactuals.

 2. There are unnecessary notations in the paper that can be easily simplified. For example, they only use $U$ and $V$ in the causal diagram in figure 1, but they hardly use them in the paper. $x_{u, v}$ can simply be $x$. All the concept level interventions can be shown using $x^{C_i = \hat{c}}$ and with similar form. They should mention this clearly in the paper if they have any specific reason to show $U$ and $V$. Beyond using the notion of counterfactual, causality also doe not play a role in this paper. Also, the name *proxy* is used for a different thing in causality literature, so it is quite confusing.

 3. For figure 1-d) $\mathcal{L}_{IN}$: It seems that the authors append a token to the factual text using the intervened concept to approximate the true counterfactual input. This concatenation can be problematic because of the conflicting concept in the input text and the generated token from the intervention. It is not clear to me why appending the counterfactual token is a good idea. If the authors are not appending a token and doing something else, it is not clear from the paper.

 4. The notation in equation 8 is ambiguous. The authors should specify what does $C_i \leftarrow c'$ indicate? If they meant intervened concept $C_i$ is appended to $\mathcal{B}(x_{u,v})$, they should use consistent notation in equation (2) or figure 1-d. Do they mean to append using a special concept token?

 5. The authors should clarify how they intervene in the intermediate layers of the transformer-based models. Did they intervene at the concept level in the CLS token of the intermediate layers? This is not clear from the paper.

6. One of my main criticism is their choice of evaluation metric. Yes, it makes sense to ensure that the fine-tuned model does not hurt the performance of the original model (and there are experiments to show that in the paper) however, correct estimation of the ITE neither proves a better explanation nor a more useful fine-tuned model. So it is not clear to me why they chose to report ITE.


 7. In many places (like table 4), they term the original BlackBox as 'Finetuned'. Can they specify the reason?

 8. In fig 4, they should use a color bar.

**Summary Of The Paper:**

This paper proposes a method to fine-tune a black-box model that is counterfactual consistent.
First, they obtain the true counterfactual input via either: (1) from the dataset explicitly by a human annotator or (2) by sampling the nearest sentence from the dataset using intervention to a specific concept. Next, they employ the Causal Proxy Model (CPM) -- which is a copy of the Black-box to (1) mimic the BlackBox for the factual input, (2) create a neural representation to be intervened to approximate the BlackBox for the counterfactual input. While simulating the counterfactuals, they use the true counterfactual text as input to predict the counterfactual output from the BlackBox. However, to mimic the counterfactual output from the BlackBox, they use either of the following strategies: (1) append a token to the factual text using the intervened concept to approximate the true counterfactual input and train CPM, (2) employing interchange intervention training (IIT) by Geiger et al. (2022) to intervene at the hidden neurons of CPM that get affected due to the intervention of the concepts.


**Summary Of The Review:**

As mentioned in the first point in the weakness, the authors generated consistent counterfactuals. However, they did not substantiate with their experiments how these counterfactuals will help in explanation. Some of the choice experiments are not well justified and may not be appropriate for the paper's goals. Overall, I disagree with the authors (1) about calling this method an explanation method , (2) calling the approach *causal* .

---

> ### Author Response · Authors · 2022-11-09
> **Official Response to Reviewer 9V3t (Part 2/2)**
>
> > Q5: *``The authors should clarify how they intervene in the intermediate layers of the transformer-based models. Did they intervene at the concept level in the $\texttt{[CLS]}$ token of the intermediate layers? This is not clear from the paper.’’*
>
> **A5**: Thanks! **We have extensive appendices that cover these issues**. We will link to them more prominently and summarize their contents more fully. Some details:
> How we chose our intervention site is discussed in detail in **Section 4.4**; it is the $\texttt{[CLS]}$ token at different transformer layers.
> We also try different intervention site size (how many dimensions) and location (which layer), and include a set of supplementary studies in **Appendix A.5** as noted in the paper.
>
> > Q6: *``One of my main criticism is their choice of evaluation metric.’’*
>
> **A6**: **We should clarify that our main evaluation metric is the $\widehat{\text{ICaCE}}_{\mathcal{N}}$** score, which we adopt from the benchmarking procedures in the CEBaB paper. **We follow the same evaluation metrics as in the CEBaB paper with the exact same training setup, and achieve much better performance.** CEBaB measures how well explainers are able to estimate counterfactual model behavior. The observation that CPMs maintain a good factual performance is a secondary observation that has potential practical significance: potentially one can deploy the explainer instead!
>
> > Q7: *``In many places (like **Table 4**), they term the original BlackBox as 'Finetuned'. Can they specify the reason?’’*
>
> **A7**: Apologies for the terminological confusion! We will address this. All the BlackBox models we aim to explain are Finetuned on a part of the CEBaB data, hence the confusion in terms. CPMs explain these Finetuned / Blackbox models.
>
> > Q8: *``In **Figure 4**, they should use a color bar.’’*
>
> **A8**: Agreed, we will make this change. Thank you for this suggestion.

---

> ### Author Response · Authors · 2022-11-09
> **Official Response to Reviewer 9V3t (Part 1/2)**
>
> We would like to thank the reviewer for their *in-depth feedback on our paper*, which leads us to be able to *better situate our work in the landscape of different explanation methods*. Below we provide a point-to-point response, which adds to our above response to the shared reviewer themes.
>
> > Q1: *``... The overall aim is to have counterfactual consistency, meaning the output should change reasonably by changing the counterfactual concept… Overall, I do not agree with the authors that this is an explanation method. They did not compare with some existing explanation strategies like TCAV, demonstrating how these counterfactual explanations serve a better purpose … ’’*
>
> **A1**: We really appreciate having this perspective on the work, since it will help us situate the ideas within the broader landscape.
>
> **CPMs belong to a class of explanation methods in which one trains explanation methods to predict changes in model behavior (denoted as the $\widehat{\text{CaCE}}$ score).**  Prominent existing explainers in this class include *S-Learner* [Kunzel et. al., 2019], *INLP* [Ravfogel et. al., 2020] and *CausalLM* [Feder et. al., 2021]. In the paper that introduced CEBaB, these methods are compared, and only the ones in our class are competitive. This likely traces to the extreme complexity of the models we need to explain – simple analytic explainers aren’t up to the task.
>
> In particular, *TCAV* [Kim et. al., 2018] is evaluated in the original CEBaB paper. It doesn’t show up in the current paper because its performance is not much better than chance in general. The CEBaB paper has a detailed accounting, in Table 10 in the appendices.
>
> The CEBaB paper also shows that numerous explainers – *TCAV* [Kim et. al., 2018], *ConceptSHAP* [Yeh et. al., 2020], the above-mentioned ones, and many others – can be cast as methods for predicting treatment effects. We will revise our introduction to bring this out, and we will make it clearer that this is indeed a particular (and non inevitable) perspective on explanation.
>
> In our paper, feature attribution methods play only a secondary role: they further reassure us that our methods for concept localization are indeed localizing concepts. In evaluating the claim that CPMs are good explainers, we rely entirely on the CEBaB benchmark.
>
> > Q2: *``There are unnecessary notations in the paper that can be easily simplified … Beyond using the notion of counterfactual, causality also doe not play a role in this paper.’’*
>
> **A2**: Thank you for this feedback! We are always keen to simplify notation where possible, and so we will look for ways to do this, following your suggestions. At the same time, we don’t want to take for granted notational conventions from the causal inference literature that will make the paper harder to understand for NLP researchers. $\textit{U}$ and $\textit{V}$ are interesting cases here. **We do definitely need those variables, since they are crucial to describing our sampling procedures.** This subtle use of $\textit{U}$ and $\textit{V}$ is highlighted in **Figure 1e**.
>
> > Q3: *``For **Figure 1d** : It seems that the authors append a token to the factual text using the intervened concept to approximate the true counterfactual input. This concatenation can be problematic because of the conflicting concept in the input text and the generated token from the intervention.’’*
>
> **A3**: Interesting perspective! **These tokens are new, randomly initialized tokens that we create just for this purpose.** During training, the model learns to approximate the correct counterfactual when exposed to one of these tokens. Thus, this token which previously had no meaning now places the model in a state of counterfactual reasoning. Manipulating this appended token at inference time allows us to simulate counterfactual model behavior. The tokens can be thought of as learned counterfactual prompts. We recently got a suggestion to actually try making the tokens actual texts like *``but make the food evaluation negative’’* or *``but imagine the service was good’’*. That is, intuitively expressed interventions. We will offer this intuition in the next version of the paper.
>
> > Q4: *``The notation in equation 8 is ambiguous.’’*
>
> **A4**: We will review our notation carefully, and make corresponding changes to the paper. In any case, we will expand our descriptions to further clarify the distinctions. In brief:
> $C_{i} \leftarrow c’$ illustrates the type of intervention we want to perform. In the case of GPT-3, we cannot specially append an artificial token as a vector to the input; instead, we have to construct a prompt to describe the intervention. We supply details about how we construct the prompt and how these incorporate information about the intervention $C_{i} \leftarrow c’$ in the **Appendix A.7** as noted in the paper.
> $t_{C_{i} \leftarrow c’}$ means the type of intervention we want to perform, expressed as a special token.

---

> > ### Comment · Reviewer_9V3t · 2022-11-30
> > **satisfied by the rebuttal**
> >
> > we are satisfied with the rebuttal, we will update the scores accordingly.

---

### Author Response · Authors · 2022-11-09
**Meta Response to All Reviewers**

We are *extremely grateful* to all our reviewers for their incisive commentary and *very useful suggestions* for making the paper more accessible to a broader audience.

**The reviews are also helping us to see how we can better foreground and situate the main findings of the paper.** In particular, we want to emphasize that the primary assessment of CPM models is the CEBaB assessment, which uses the same protocols as the original CEBaB paper [Abraham et. al., 2022; cite once to avoid redundancy]. CEBaB measures how well an explainer estimates the effect of input-level concept changes on a model’s output. Thus, good CEBaB performance entails a good counterfactual explanation of the model behavior.

**We show that CPMs beat all 7 methods evaluated in that paper by wide margins** including *Approx* [Abraham et. al., 2022], *CONEXP* [Goyal et. al., 2020], *S-Learner* [Kunzel et. al., 2019], *TCAV* [Kim et. al., 2018], *ConceptSHAP* [Yeh et. al., 2020], *INLP* [Ravfogel et. al., 2020] and *CausalLM* [Feder et. al., 2021], and CPMs even beat our improved version of the best of those methods (S-Learner). CEBaB seems to us to be the best available benchmark for all these methods; very little needs to be done to apply them to the CEBaB task.

Our paper offers additional assessments and observations: **$\textbf{CPM}_{\textbf{HI}}$ models do indeed localize information** in the way we expect (as shown by our Integrated Gradients based analysis), and **CPMs are good at the factual task**, in addition to **being good explainers**. These are important but secondary experiments.

Additionally, we want to emphasize that **CPMs are not especially demanding when it comes to the needed causal graph or associated data**. The causal graph can be very partial, and our approximate counterfactual strategies require metadata of a sort that is widely available on the Web. CEBaB itself has much more specialized and rare data, but that is because it is an evaluation tool for explainers.

We also want to elaborate on a point of ambiguity that several reviews helped us surface. **$\text{CPM}_{\textbf{IN}}$** takes as input a token $t_{C_{i} \leftarrow c’}$ carrying information about the concept-level intervention. We randomly initialize these special tokens and learn their embedding during training. We will update our manuscript to clarify this point.

In conclusion, the reviews helped us better situate the main contributions of our work. Towards the end of the review process, an updated version of our paper incorporating the resulting changes will be made available before the Discussion Stage 1 deadline.

---

### Author Response · Authors · 2022-11-15
**Additional Meta Response with Rebuttal Revision**

We would like to thank the reviewers again for their valuable comments, which have led us to make important revisions to the paper, particularly by supplying new baseline results with X-Learner [Kunzel et al., 2019] and showing CPMs outperforms X-Learner by a large margin. We have also uploaded a revised paper with **changes marked in red**.

**Overview of major changes (now appearing in the rebuttal revision):**
- **Figure 1 (a)**: To remain aligned with the CEBaB paper, we have made only a minor change for now to **Figure 1 (a)**, to clarify that the star-rating acts are conceptually included in the exogenous variables.
- **Section 3**: clarifying the training objective of **$\textbf{CPM}_\textbf{IN}$** and **$\textbf{CPM}_\textbf{HI}$**. We also clarify the definition of **$t_{C_{i} \leftarrow c’}$**.
- **Section 4.3**: clarifying the fact that we are outperforming 7 methods evaluated in that paper by wide margins including *Approx* [Abraham et. al., 2022], *CONEXP* [Goyal et. al., 2020], *S-Learner* [Kunzel et. al., 2019], *TCAV* [Kim et. al., 2018], *ConceptSHAP* [Yeh et. al., 2020], *INLP* [Ravfogel et. al., 2020] and *CausalLM* [Feder et. al., 2021]. These results are aggregated to a single column in our main result Table 1. We also clarify the setup of two new strongest baselines *S-Learner* and *GPT-3*.
- **Section 6**: clarifying the fact that CPMs do not require a full causal graph behind the dataset.
Appendix A.4: Adding implementation details about *X-Learner* along with results in Table 7. *X-Learner* achieves performance similar to *S-Learner*. CPMs outperform *X-Learner* by large margins.

We also made other minor changes for clarification purposes with **changes marked in red**.

---

### Decision · Program_Chairs · 2023-01-20

**Decision:**

Reject

**Justification For Why Not Higher Score:**

Not good enough

**Justification For Why Not Lower Score:**

N/A

**Metareview: Summary, Strengths And Weaknesses:**

This paper proposes Causal Proxy Model or CPM to causally explain black-box models. It proposes to collect (approximate) counterfactuals (via crowd and nearest search) and do counterfactual consistency training. The method shows promising results but reviewers expressed their concerns regarding: (a) its motivation (9V3t, aAEJ), (b) counterfactual generation and its novelty (MZ4t), (c) method being specific to the CEBaB dataset and (d) presentation issues (multiple reviewers). Thanks to the authors for clarifying many points during the rebuttal. Despite this, the reviewers were not entirely convinced in the post-rebuttal discussion. I hope the authors can fix these issues in their next version.